# Evolutionary System Design with Answer Set Programming

Christian Haubelt [1], Luise Müller [1], Kai Neubauer [2], Torsten Schaub [3,]*, and Philipp Wanko [3,]*

1 Applied Microelectronics and Computer Engineering, University of Rostock, 18051 Rostock, Germany
2 Bosch Sensortec GmbH, 01109 Dresden, Germany
3 Department of Computer Science, University of Potsdam, 14482 Potsdam, Germany
* Correspondence: torsten@cs.uni-potsdam.de (T.S.); wanko@cs.uni-potsdam.de (P.W.)

**Abstract:** We address the problem of evolutionary system design (ESD) by means of answer set programming modulo difference constraints (AMT). The goal of this design approach is to synthesize new product variants or generations from existing products. We start by formalizing the underlying system synthesis problem and design space exploration process, which consists of finding the Pareto front with respect to latency, cost, energy, and similarity measures between the two designs. We then present AMT-based encodings to capture all of these aspects. The idea is to use plain ASP for conflict detection and resolution and for routing and to use difference constraints for scheduling. Moreover, we propose a new approach for expressing the similarity that we use at three alternative levels of AMT-based design space exploration, namely, at the strategic, heuristic, and objective levels, which is performed to guide the exploration towards designs of high interest. Last but not least, we systematically evaluate the emerging techniques empirically and identify the most promising AMT techniques.

**Keywords:** evolutionary system design; design space exploration; answer set programming modulo theories; difference constraints

## 1. Introduction

Application-specific computer systems are the backbone of any smart product today; such computers are usually referred to as embedded systems, as they are embedded in a larger product. In fact, in this context, the number of embedded systems exceeds the number of general-purpose computers by several orders of magnitude. Because of their integration into larger products and their close interaction with their physical environment, embedded systems must meet stringent design constraints, including size/volume, timing, power, energy, etc. In addition, the design of an embedded system must meet extremely tight time-to-market requirements. All of this continues to be true for embedded systems, which are becoming increasingly more complex. This growth in complexity is due to both more sophisticated applications and more sophisticated computing platforms. In summary, the design of embedded systems is a complex task that must be approached in a process that is as automated as possible.

As a consequence, implementations of embedded systems are derived from their specifications. This step is called synthesis and involves a vast number of individual but often interdependent design decisions. These decisions include: (a) the selection of processors, hardware accelerators, memories, and communication infrastructures; (b) the distribution of functions to computational cores, distribution of variables to memories, and the routing of memory transactions through the communication infrastructure; and (c) the scheduling of computations, memory accesses, and memory transactions. Finding a feasible solution for all dependent design decisions is a hard problem in itself; however, in addition, the final product must also be optimized, which leads to a multi-objective combinatorial optimization problem. This step in the design process is called design

space exploration (DSE), which involves the identification of optimal solutions or at least promising candidates.

In the past, we have shown that answer set programming modulo theories [1], or AMT for short, outperform other approaches for DSE in communication-intensive embedded systems [2]. The general idea is to use plain ASP for conflict detection and resolution and for routing; furthermore, difference constraints are used for scheduling. However, its applicability is still limited to relatively small systems.

In this paper, we explore how the knowledge of a previous version of an embedded system can be used in an AMT-based DSE to find good design candidates faster, thus extending the applicability of this approach. Our approach is motivated by the observation that hardly any complex system is designed from scratch but is instead derived and extended from a previous version. In the literature, this is referred to as product generation engineering (PGE; [3]). Even in the case of new product developments from scratch, numerous variants are often derived from a reference configuration, thus forming an entire product line; both scenarios can be regarded as evolutionary system design (ESD). As a consequence, ESD leads to design approaches where new product variants or generations are derived from existing products.

To keep the time-to-market low, the best strategy is often to leave parts that are needed in several versions of the system unchanged, i.e., to make identical design decisions as often as possible. To achieve the goal of retaining design decisions as much as possible, the similarity between different system implementations must be formally defined, even when specifications have been (slightly) changed. In a previous work, we have explored this issue and shown how to use an initially proposed similarity measure in an ad hoc AMT-based DSE [4]. Based on the first promising findings, we now systematically explore this topic in greater depth with a particular focus on the impact of alternative ASP techniques. Moreover, we take a new approach for expressing similarity and use it at three different levels of AMT-based DSE, namely, (a) as a strategy, (b) as a heuristic, and (c) as an objective, to guide the exploration towards the regions of high interest.

To begin with, in Section 2, we provide a brief introduction to ASP by focusing on its modeling language. We then formalize all aspects of ESD in Section 3, namely, the system synthesis problem along with the DSE process, which consists of finding the Pareto front regarding the quality measures of latency, cost, and energy consumption, and similarity measures between two implementations. Sections 4 and 5 detail our AMT-based approach to system design and DSE, respectively. The former presents the general problem and how binding, routing, and scheduling are addressed in AMT; furthermore, we describe how multi-objective optimization is accomplished. The latter section deals with the encoding of ESD in AMT, focusing on the three aforementioned alternatives for addressing similarities: strategies, preferences, and (domain-specific) heuristics. Finally, Section 6 reports on an extensive empirical evaluation, which contrasts 85 different setups, and the evaluation identifies the most promising combinations of AMT techniques for ESD. We summarize our approach in Section 7.

## 2. Answer Set Programming

A logic program consists of rules of the form:

$$\texttt{a}_1\texttt{;}\ldots\texttt{;}\,\texttt{a}_m \quad \texttt{:-}\quad \texttt{a}_{m+1}\texttt{,}\ldots\texttt{,}\,\texttt{a}_n\texttt{,}\,\texttt{not}\ \ \texttt{a}_{n+1}\texttt{,}\ldots\texttt{,}\,\texttt{not}\ \ \texttt{a}_o$$

where each $\texttt{a}_i$ is an atom of form $\texttt{p}(\texttt{t}_1,\ldots,\texttt{t}_k)$ and all $\texttt{t}_i$ are terms, which are composed of function symbols and variables. For $1 \leq m \leq n \leq o$, atoms $\texttt{a}_1$ to $\texttt{a}_m$ are often called head atoms, while $\texttt{a}_{m+1}$ to $\texttt{a}_n$ and $\texttt{not}\ \texttt{a}_{n+1}$ to $\texttt{not}\ \texttt{a}_o$ are also referred to as positive and negative body literals, respectively. An expression is said to be ground if it contains no variables. As usual, $\texttt{not}$ denotes (default) negation. A rule is called a fact if $m = n = o = 1$, normal if $m = 1$, and an integrity constraint if $m = 0$. In what follows, we only deal with normal logic programs, for which $m$ is either 0 or 1. Semantically, a logic program induces a set of stable models, which are distinguished models of the program as determined by the stable models' semantics [5].

To ease the use of ASP in practice, several extensions have been developed. First of all, rules with variables are viewed as shorthands for the set of their ground instances. Further language constructs include conditional literals and cardinality constraints [6]. The former items are of the form $a:b_1,\ldots,b_m$, (In rule bodies, they are terminated by ';' or '.' [7]); the latter can be written as $s\,\{d_1;\ldots;d_n\}\,t$ (More elaborate forms of aggregates are obtained by explicitly using function (e.g., `#count`) and relation symbols (e.g., `<=`) [7]), where $a$ and $b_i$ are possibly negated (regular) literals and each $d_j$ is a conditional literal. $s$ and $t$ provide optional lower and upper bounds on the number of satisfied literals in the cardinality constraint. We refer to $b_1,\ldots,b_m$ as a condition. The practical value of both constructs becomes apparent when used with variables. For instance, a conditional literal such as `a(X):b(X)` in a rule's body expands to the conjunction of all instances of `a(X)` for which the corresponding instance of `b(X)` holds. Similarly, `2 {a(X):b(X)} 4` is true whenever at least two and at most four instances of `a(X)` (subject to `b(X)`) are true; more sophisticated examples are given in Section 4.

A particular convenience feature is anonymous variables, which are uniformly denoted by an underscore '_'. Each underscore in a rule is interpreted as a fresh variable. In turn, atoms with anonymous variables are replaced by new atoms dropping these variables; the new atoms are then linked to the original ones by rules expressing projections. For instance, an atom such as `task(T,_)` is replaced by `task'(T)` while adding the rule `task'(T) :- task(T,X)`.

As an example, consider the rule in Line 1 of Listing 2:

```
1 { bind(T,R) : mapping(R,T) } 1 :- task(T).
```

This rule has a single head atom consisting of a cardinality constraint; it comprises all instances of `bind(T,R)`, where `T` is constrained by the single body literals and `R` varies over all instantiations of predicates `mapping/2`. Given 12 resources, this results in 12 instances of `bind(R,T)` for each valid replacement of `T`, among which exactly one must be chosen according to the above rule.

Finally, let us consider some system directives particular to *clingo* (solver directives are preceded with a hash symbol in *clingo* [7]).

To begin with, objective functions minimizing the sum over the first numeral argument $w_i$ of a set of weighted tuples $(w_i,t_{1_i},\ldots,t_{k_i})$, whose membership is subject to condition $b_{1_i},\ldots,b_{l_i}$, are expressed as

```
#minimize{w_1@l_1,t_1_1,...,t_k_1:b_1_1,...,b_l_1;...;w_n@l_n,t_1_n,...,t_k_n:b_1_n,...,b_l_n}.
```

Lexicographically ordered objective functions are (optionally) distinguished via levels that are indicated by $l_i$. An omitted level defaults to zero.

Furthermore, *clingo* offers means for manipulating the solver's decision heuristics. Such heuristic directives are of the form

```
#heuristic a:b_1,...,b_m. [w,m]
```

where $a:b_1,\ldots,b_m$ is a conditional literal; `w` is a numeral term; and `m` a heuristic modifier, indicating how the solver's heuristic treatment of `a` should be changed whenever $b_1,\ldots,b_m$ holds. Whenever `a` is chosen by the solver, `sign` enforces that it becomes either true or false depending on whether $w$ is positive or negative, respectively. The modifier `level` partitions all atoms in focus according to the given weight and then selects atoms with decreasing weight. Finally, the modifiers `true` and `false` constitute a combination of `sign` and `level`. See [7,8] for a comprehensive introduction to heuristic modifiers in *clingo*.

Furthermore, *clingo* features an integrated acyclicity checker. Acyclicity constraints are expressed by edge directives in the form

```
#edge (u,v):b_1,...,b_m.
```

where `u` and `v` are terms representing an edge from node `u` to node `v` and $b_1,\ldots,b_m$ is a condition. The arc `(u,v)` belongs to a (internal) graph whenever the condition holds. Once such directives are present, a stable model is only output by *clingo* if its induced graph is acyclic [9].

In fact, in this paper we rely on the extension of *clingo* with difference constraints, viz. *clingo*[DL]. Difference constraints are expressed as theory atoms having the form (theory atoms are preceded with an ampersand in *clingo* [1])

```
&diff { u-v } <= d
```

where `u` and `v` are terms and `d` is a numeral term; they may occur as head atoms or body literals. Each such theory atom is associated with a difference constraint $u - v \leq d$, where $u, v$ are integer variables and $d$ is an integer. In this setting, a stable model is only obtained if the set of difference constraints associated with the theory atoms in the stable model is satisfiable [10]. In *clingo*[DL], the obtained integer assignment is captured by expressions using predicate `dl/2`. For instance, the assignment $u \mapsto 3$ is output as `dl(u,3)`. In fact, the satisfaction of a set of difference constraints can be reduced to an acyclicity check of a weighted graph, where each difference constraint $u - v \leq d$ induces an edge from node $u$ to node $v$, weighted with $d$. Whenever a cycle is present whose sum of weights is negative, the set of difference constraints is unsatisfiable. In view of this, difference constraints can be seen as an extension of acyclicity constraints with distances.

Full details on the input language of *clingo* along with various examples can be found in the *Potassco User Guide* [7].

## 3. Evolutionary Design Space Exploration

We characterize evolutionary design space exploration in three steps. First, we define the system synthesis problems that captures a set of applications that have to be executed on a hardware platform. Then, we identify desirable implementations with respect to multiple objective values. Finally, we use distance metrics to identify similar solutions. We illustrate how we can exploit the similarity to high quality implementations to identify good implementations to different but similar system synthesis problems.

### 3.1. System Synthesis Problem

Given a set $\mathbb{T}$ of tasks and set $\mathbb{D}$ of dependencies among them, an *application* is a directed acyclic graph $(T, D)$ consisting of tasks $T \subseteq \mathbb{T}$ and dependencies $D \subseteq \mathbb{D}$ among tasks.

A *hardware platform* is a tuple $((R, L), rd, c, se, re, p, m, e, de)$ where:

- $(R, L)$ is a directed graph consisting of resources $R$, i.e., either routers, computational, or memory resources, and links $L$ between resources describing the network;
- $rd \in \mathbb{N}$ is the (uniform) routing delay, i.e., the time it takes for a communication to traverse a link;
- $c : R \to \mathbb{N}$ is a total function that gives the cost of a resource;
- $se : R \to \mathbb{N}$ is a total function that gives the static energy consumption of a resource;
- $re \in \mathbb{N}$ is the (uniform) routing energy, i.e., the energy used whenever a link is used by a communication;
- $m : R \to 2^{\mathbb{T}}$ is a total function that gives the set of tasks that may be executed on a resource;
- $p \in \mathbb{N}$ is the (uniform) period in which applications have to be executed, i.e., the deadline of all tasks;
- $e : R \times \mathbb{T} \to \mathbb{N}$ is a partial function that gives the execution time of tasks; that is, $e(r, t)$ is defined if $t \in m(r)$ for resource $r \in R$ and task $t \in \mathbb{T}$;
- $de : R \times \mathbb{T} \to \mathbb{N}$ is a partial function giving the dynamic energy consumption that is dependent on what specific task is executed on what resource; as above, $e(r, t)$ is defined if $t \in m(r)$ for resource $r \in R$ and task $t \in \mathbb{T}$.

A *system synthesis problem* is a pair $(A, P)$ consisting of a set $A$ of applications and a hardware platform $P$. We assume that no two applications share a task, that is, $\bigcap_{(T,D) \in A} T = \varnothing$.

For illustration, we provide a small example system synthesis problem in Figure 1.

On the left, we give the set of applications, which, in this case, is a single dependency graph comprising four tasks $t_1, t_2, t_3$, and $t_4$. More precisely, the (singleton) set of applications is

$$A = \{(\{t_1, t_2, t_3, t_4\}, \{(t_1, t_2), (t_1, t_3), (t_2, t_4), (t_3, t_4)\})\}$$

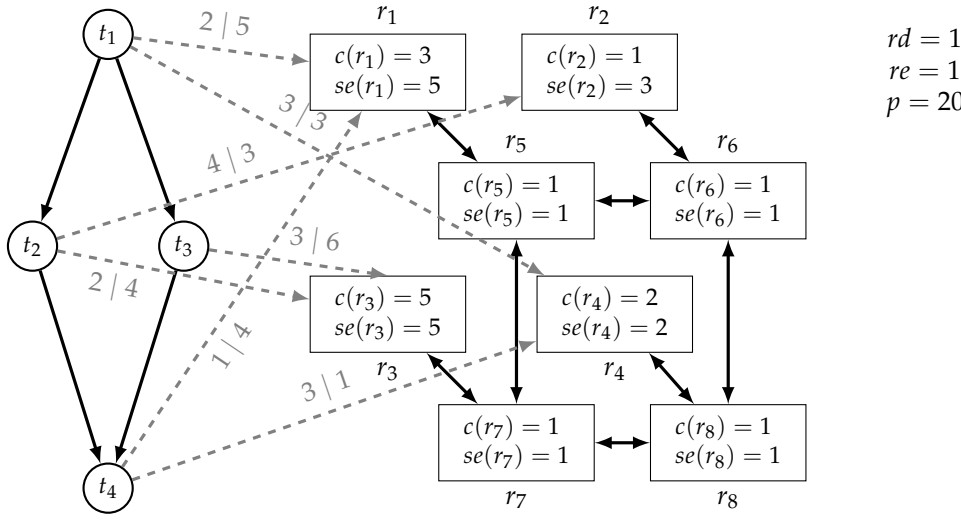

**Figure 1.** Example system synthesis problem (parent specification).

On the right, we have the hardware platform with the four computational resources $r_1$ to $r_4$, and four routers $r_5$ to $r_8$. The architecture follows a grid-like network-on-chip (NoC [11]) structure. This is not necessary for our formalization but is usually assumed and present in our benchmark instances. Each computational resource $r_i$ is connected to a router $r_{i+4}$ in both directions for $i \in \{1, 2, 3, 4\}$. Furthermore, routers are connected via edges:

$$\{(r_5, r_6), (r_5, r_7), (r_6, r_5), (r_7, r_5), (r_8, r_6), (r_8, r_7), (r_6, r_8), (r_7, r_8)\}$$

All uniform characteristics, such as routing delay *rd*, routing energy *re*, and period *p*, are given in the upper right corner and are set to one, one, and twenty, respectively. The cost and static energy consumption of the resources are given inside the rectangles labeled with the resource names. All routers have a uniform cost and energy consumption of one, i.e., they are all the same type of hardware component. The computational resources have the following attributes:

$$c(r_1) = 3, \qquad se(r_1) = 5, \qquad c(r_2) = 1, \qquad se(r_2) = 3,$$
$$c(r_3) = 5, \qquad se(r_3) = 5, \qquad c(r_4) = 2, \qquad se(r_4) = 2.$$

The mapping function *m* is given by the dashed gray arrows from the task nodes to the resources that they can be executed on. The first part of the label is the execution time on the resource, and the second part is the dynamic energy consumption. In detail, we have the mapping options $m(r_1) = \{t_1, t_4\}, m(r_2) = \{t_2\}, m(r_3) = \{t_2, t_3\}, m(r_4) = \{t_1, t_4\}$, with execution times *e* and dynamic energy consumption *de*:

| $e:$ | $r_1$ | $r_2$ | $r_3$ | $r_4$ |
|---|---|---|---|---|
| $t_1$ | 2 | – | – | 3 |
| $t_2$ | – | 4 | 2 | – |
| $t_3$ | – | – | 3 | – |
| $t_4$ | 1 | – | – | 3 |

| $de:$ | $r_1$ | $r_2$ | $r_3$ | $r_4$ |
|---|---|---|---|---|
| $t_1$ | 5 | – | – | 3 |
| $t_2$ | – | 3 | 4 | – |
| $t_3$ | – | – | 6 | – |
| $t_4$ | 4 | – | – | 1 |

Essentially, $r_1$ and $r_3$ can be seen as slightly more costly and energy-intensive resources with faster execution times, while $r_2$ and $r_4$ are less expensive but slower.

### 3.2. Implementations of a System Synthesis Problem

From here on, we consider a set of applications $A$ to determine our basic sets for task and dependencies. That is, we have $\mathbb{T} = \bigcup_{(T,D) \in A} T$ and $\mathbb{D} = \bigcup_{(T,D) \in A} D$.

Furthermore, consider a hardware platform with resources $R$. A *binding* is a total function $\mathbb{T} \to R$ that assigns each task a resource that it is executed on.

Let $\mathcal{R} = \{(r_i)_{i=1}^n \mid \{r_1, \dots, r_n\} \subseteq R, n \in \mathbb{N}\}$ be the set of subsequences of elements of $R$. We sometimes abuse set notation and write $r \in S$, $(r, r') \in S$, or $S \cup S'$ for $r \in R$, $(r, r') \in L$, and $\{S, S'\} \subseteq \mathcal{R}$ to denote that resource $r$ or link $(r, r')$ are part of a sequence $S$ or refer to the union of all resources occurring in $S$ and $S'$, respectively. We define the length of such a sequence as $|(r_i)_{i=1}^n| = n - 1$.

A *routing* is a total function $\mathbb{D} \to \mathcal{R}$ that assigns each dependency a sequence of resources that stand for the route of the communication between the two tasks.

A *scheduling* is a total function $\mathbb{T} \cup \mathbb{D} \to \mathbb{N}$ that assigns each task and dependency a starting time, i.e., when the tasks start executing and the communication between two tasks is initiated.

A binding $b$ is *valid* if $t \in m(b(t))$ for all $t \in \mathbb{T}$. That is, we can only bind tasks to resources on which they may be executed.

Given a valid binding $b$, a routing $r$ is *valid* on a hardware platform with links $L$ if the following conditions are satisfied:

1. if $r(d) = (\dots, r_i, r_{i+1}, \dots)$, then $(r_i, r_{i+1}) \in L$ for all $d \in \mathbb{D}, i \in \mathbb{N}$;
2. if $b(t) = r = b(t')$, then $r((t, t')) = ()$ for $(t, t') \in \mathbb{D}$;
3. if $b(t) = r \neq b(t') = r'$, then $r((t, t')) = (r, \dots, r')$ for $(t, t') \in \mathbb{D}$.

Condition 1 ensures that the routing respects the hardware architecture in that every adjacent resource in a route has to be a link in the network graph. Condition 2 enforces an empty sequence if two depending tasks are mapped to the same resource. Condition 3 requires that every route respects the binding, i.e., it starts on the resource with the sending task and ends at the resource with the receiving task, whenever tasks are bound to different resources.

Given a valid binding $b$ and a valid routing $r$, a scheduling $s$ is *valid* on a hardware platform with resources $R$ if the following conditions are satisfied:

1. $0 \leq s(t) \leq s(t) + e(b(t), t) \leq p$ for $t \in \mathbb{T}$;
2. $s(t) + e(b(t), t) \leq s((t, t'))$ for $(t, t') \in \mathbb{D}$;
3. $s((t, t')) + |r((t, t'))| * rd \leq s(t')$ for $(t, t') \in \mathbb{D}$;
4. if $b(t) = b(t')$, then for $t \neq t'$ and $\{t, t'\} \subseteq \mathbb{T}$, either

    (a) $s(t) + e(b(t), t) \leq s(t')$; or
    (b) $s(t') + e(b(t'), t') \leq s(t)$;

5. if $r(d) = (\dots, r, r', \dots)$ and $r(d') = (\dots, r, r', \dots)$, then for $d \neq d'$, $\{d, d'\} \subseteq \mathbb{D}$ and $\{r, r'\} \subseteq R$, either

    (a) $s(d) + |r(d)| * rd \leq s(d')$; or
    (b) $s(d') + |r(d')| * rd \leq s(d)$.

Condition 1 enforces that every task's start time is at least zero, and it is chosen in a way that it finishes before the period. Condition 2 states that communications may only start after the task that sends them has finished. Note that both conditions depend on the binding $b$ as the execution time may be different for different resources. Condition 3 handles dependencies between tasks in that a receiving task may only start once all tasks it depends upon have finished executing and their messages have been received. Note that in the case that both tasks in a dependency are mapped to the same resource, we have $|r((t, t'))|$ equal zero, so only the start plus the execution time of the first task delays the second task without any communication overhead. Whenever two resources are bound to

the same resource, Condition 4 ensures that their execution does not overlap; either the first task has to finish before the second may start, or vice versa. Note that in case both tasks depend on each other, satisfaction of this condition follows from Condition 3. Furthermore, in our formalization, every resource can only execute one task at a time. Finally, whenever two routes share a link, Condition 5 requires either communication to finish before the other may start. This is called a *circuit switching strategy* [11] because the message blocks the whole route until it is received.

A triple $(b, r, s)$ is an implementation of a system synthesis problem $(A, P)$ if $b$ is a valid binding, $r$ is a valid routing with respect to $b$, and $s$ is a valid scheduling with respect to $b$ and $r$.

The small example system synthesis problem in Figure 1 already has 17,056 implementations. When only considering schedules that are executed as early as possible, i.e., tasks are executed as soon as they have received the necessary communications, the resources they are bound to are free, communications are sent as soon as the sender task is finished, and the communication's route is free, we obtain 62 implementations.

One of these implementations is given in Figure 2. In Figure 2a, binding is represented by red arrows that select for every task an appropriate resource. In detail, we have $b(t_1) = r_1, b(t_2) = r_3, b(t_3) = r_3$, and $b(t_4) = r_1$. Note that unused links and resources are grayed out as they are not needed for this particular implementation. Routing and scheduling are shown in Figure 2b. The x-axis of the graph represents time units up to the period of 20. The y-axis are resources and links used for binding tasks and communications, respectively. Which links are used for which communication can be seen with the label of the dependencies at the red rectangles, and the order is given from left to right. More precisely, $r((t_1, t_3)) = (r_1, r_5, r_7, r_3) = r((t_1, t_2))$ and $r((t_3, t_4)) = (r_3, r_7, r_5, r_1) = r((t_2, t_4))$. The position and length of the red rectangles labeled with task names and dependencies on the x-axis indicate the start time and duration of execution, respectively. While a communication on a link uniformly takes one time unit, the durations of the tasks vary depending on the resources that they are bound to. The precise scheduling $s$ is as follows: $s(t_1) = 0, s((t_1, t_3)) = 2, s(t_3) = 5, s((t_1, t_3)) = 5, s(t_2) = 8, s((t_3, t_4)) = 8, s((t_2, t_4)) = 11$, and $s(t_4) = 14$.

### 3.3. Implementation Quality and Pareto Front

We evaluate an implementation of a system synthesis problem via three objective functions: cost, energy consumption, and latency.

Given an implementation $(b, r, s)$ of a system synthesis problem $(A, ((R, L), rd, c, se, re, p, m, e, de))$, we define the following total functions:

- *Cost*

$$f_c(b, r) = \sum_{u \in \{b(t) | t \in \mathbb{T}\} \cup \bigcup_{d \in \mathbb{D}} r(d)} c(u)$$

- *Energy consumption*

$$f_e(b, r) = \sum_{u \in \{b(t) | t \in \mathbb{T}\} \cup \bigcup_{d \in \mathbb{D}} r(d)} se(u) + \sum_{t \in \mathbb{T}} de(b(t), t) + \sum_{d \in \mathbb{D}} (|r(d)|) * re$$

- *Latency*

$$f_l(s) = \max\{s(t) + e(b(t), t) \mid t \in \mathbb{T}\}$$

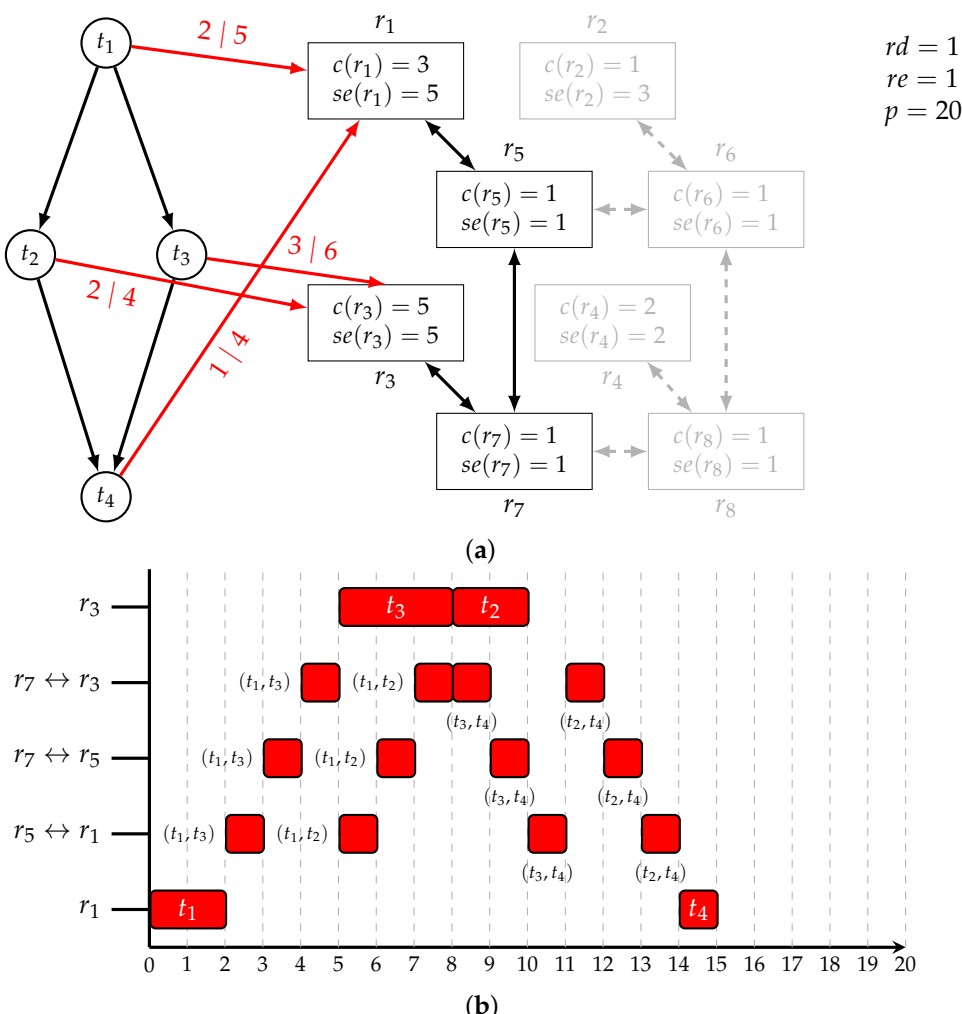

**Figure 2.** Example (parent) implementation. (**a**) Binding; (**b**) routing and scheduling.

Strictly speaking, all cost functions also depend on the system synthesis problem; however, we refrain from making this explicit for the sake of simplicity.

The function $f_c$ sums up the cost of all resources that are used, either by having a task bound to it or by being used as a routing device in a communication. The cost of our example implementation in Figure 2 is 10 as we do not use $r_2, r_4, r_6$, or $r_8$. The energy consumption $f_e$ considers the static energy required by using a resource, the dynamic energy stemming from the execution of a task on a resource, and the energy consumption of the links used in any communication. Note that, in contrast to the static energy of a resource, the energy consumption of a link is counted per use in a communication. The implementation in Figure 2 has an energy consumption of 43. Finally, latency $f_l$ calculates the maximum of starting plus the execution time of all tasks, i.e., the time span of the entire execution of the implementation. The example implementation in Figure 2 has a latency of 10, which is the moment that $t_4$ finishes and the application has been executed.

The *quality* of an implementation $(b, r, s)$ of a system synthesis problem $(A, P)$ is given by the tuple $(f_c(b), f_e(b, r), f_l(s))$. Accordingly, the quality of the implementation in Figure 2 is $(10, 43, 15)$.

Rather than one optimal implementation, we are interested in a set of non-dominated implementations, the so-called *Pareto front* [12]. Let $(b, r, s)$ and $(b', r', s')$ be two implementations of a system synthesis problem $(A, P)$ and $(q_1, q_2, q_3)$ and $(q'_1, q'_2, q'_3)$ be their respective quality; then, $(b, r, s)$ *dominates* $(b', r', s')$, denoted as $(b, r, s) \prec (b', r', s')$, if

1. $q_i \leq q'_i$ for all $i \in \{1, 2, 3\}$; and
2. $q_i < q'_i$ for some $i \in \{1, 2, 3\}$.

Note that $\prec$ forms a partial relation, i.e., the qualities of some implementations are incomparable. Intuitively, the quality of an implementation has to be at least as good in all aspects and strictly better in one to dominate another implementation's quality. Then, an implementation $(b, r, s)$ of a system synthesis problem $(A, P)$ belongs to the Pareto front if no other implementation $(b', r', s')$ of $(A, P)$ exists, such that $(b', r', s') \prec (b, r, s)$.

The system synthesis problem in Figure 1 has four non-dominated implementations; two of them bind $t_1$ and $t_4$ to $r_1$ and $t_2$ and $t_3$ to $r_3$, and the remaining two bind $t_1$ and $t_4$ to $r_4$ and $t_2$ and $t_3$ to $r_3$. The two respective implementations have the same quality, namely, $(10, 43, 15)$ and $(9, 35, 16)$, respectively, as well as identical routing and binding; the sole difference is whether $t_2$ or $t_3$ is executed first. We can identify two optimal trade-offs here. The implementations with quality $(10, 43, 15)$ emphasize a shorter latency with higher cost and energy consumption, and implementations with quality $(9, 35, 16)$ save costs and energy with a slightly longer execution time. The implementation in Figure 2 belongs to the Pareto front and is one of the two non-dominated implementations favoring latency.

### 3.4. Distance between Implementations

As mentioned above, we are using distance information to foster high-quality implementations for novel but similar system synthesis problems. To this end, we start from a system synthesis problem wherein a high-quality implementation is known; this is called the *parent specification* and *parent implementation*, respectively. In our example, Figure 1 constitutes the parent specification and Figure 2 constitutes the parent implementation. Then, the parent specification is changed or updated to a so-called *child specification*.

Such a child specification for our example is given in Figure 3. Here, task $t_1$ was deleted, which can be seen as a software update as this task is no longer needed to execute the application; updates to the hardware platform are marked in green. Essentially, we replace resource $r_1$ with a more costly version that saves energy. Everything else remains unchanged from the parent specification in Figure 1.

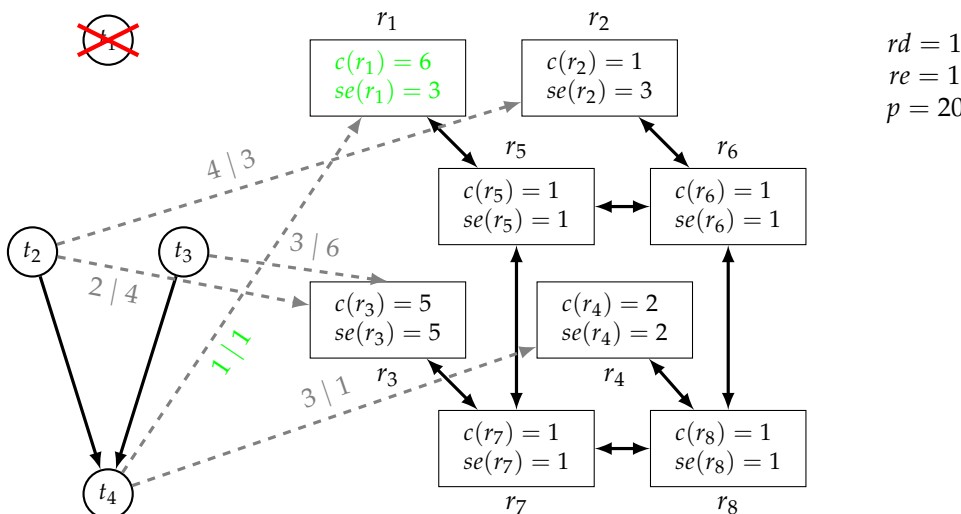

**Figure 3.** Example system synthesis problem (child specification).

Next, we define a distance measure that allows us to compare implementations. We divide the distance between two implementations of a system synthesis problem into the distance between binding, routing, and scheduling. Before accomplishing this, we require some intermediate definitions.

Given a system synthesis problem $(A, ((R, L), rd, c, se, re, p, m, e, de))$, we define the sets:

$$M_m = \{(t, r) \mid t \in m(r), r \in R, t \in T\}$$

$$L_{A,L} = \bigcup_{(T,D) \in A} \{(d, l) \mid d \in D, l \in L\}$$

$$T_A = \bigcup_{(T,D) \in A} \{t \mid t \in T \cup D\}$$

Intuitively, the set $M_m$ contains all possible bindings given a mapping function; $L_{A,L}$ contains all possible pairs of dependencies and links to collect which communication routes are possible; finally, $T_A$ is the set of all entities that are scheduled, i.e., all tasks and communications.

Given two system synthesis problems $(A, ((R, L), rd, c, se, re, p, m, e, de))$ and $(A', ((R', L'), rd', c', se', re', p', m', e', de'))$, we define three functions:

- For two bindings $b$ and $b'$ and $(t, r) \in M_m \cup M'_m$, we define

$$d_b(M_m, M_{m'}, b, b', (t, r)) = \begin{cases} 1, & \text{if } (t, r) \notin M_m, (t, r) \in M_{m'}, b'(t) = r \\ 1, & \text{if } (t, r) \in M_m, (t, r) \notin M_{m'}, b(t) = r \\ 1, & \text{if } (t, r) \in M_m, (t, r) \in M_{m'}, b(t) = r, b'(t) \neq r \\ 1, & \text{if } (t, r) \in M_m, (t, r) \in M_{m'}, b(t) \neq r, b'(t) = r \\ 0, & \text{otherwise} \end{cases}$$

- For two routings $r$ and $r'$ and $(d, l) \in L_{A,L} \cup L_{A',L'}$, we define

$$d_r(L_{A,L}, L_{A',L'}, r, r', (d, l)) = \begin{cases} 1, & \text{if } (d, l) \notin L_{A,L}, (d, l) \in L_{A',L'}, l \in r'(d) \\ 1, & \text{if } (d, l) \in L_{A,L}, (d, l) \notin L_{A',L'}, l \in r(d) \\ 1, & \text{if } (d, l) \in L_{A,L}, (d, l) \in L_{A',L'}, l \in r(d), l \notin r'(d) \\ 1, & \text{if } (d, l) \in L_{A,L}, (d, l) \in L_{A',L'}, l \notin r(d), l \in r'(d) \\ 0, & \text{otherwise} \end{cases}$$

- For two schedulings $s$ and $s'$ and $t \in T_A \cup T_{A'}$, we define

$$d_s(T_A, T_{A'}, s, s', t) = \begin{cases} 1, & \text{if } t \notin T_A, t \in T_{A'} \\ 1, & \text{if } t \in T_A, t \notin T_{A'} \\ 1, & \text{if } t \in T_A, t \in T_{A'}, s(t) \neq s'(t) \\ 0, & \text{otherwise} \end{cases}$$

These four functions determine one specific possible mapping, communication link, and start time, and whether a different decision has been made in two bindings, routings, and schedulings, respectively. The multiple cases stem from the possibility that both system synthesis problems are different. We have to take into account that certain possibilities do not exist in one or the other system synthesis problem. In case an option is unavailable but used by the other implementation, it should constitute a difference. If the missing option is not employed, the implementations are equal with respect to that option.

Given two implementations $(b, r, s)$ and $(b', r', s')$ of two system synthesis problems $(A, P)$ and $(A', P')$, respectively, we define the following distance functions:

- *Binding distance*

$$D_b(m, m', b, b') = \sum_{(t,r) \in M_m \cup M_{m'}} d_b(M_m, M_{m'}, b, b', (t, r))$$

- *Routing distance*

$$D_r(A, A', L, L', r, r') = \sum_{(d,l) \in L_{A,L} \cup L_{A',L'}} d_r(L_{A,L}, L_{A',L'}, r, r', (d, l))$$

- *Scheduling distance*

$$D_s(A, A', s, s') = \sum_{t \in T_A \cup T_{A'}} d_s(T_A, T_{A'}, r, r', t)$$

- *Overall distance*

$$D((A, P), (r, b, s), (A', P'), (r', b', s')) = D_b(m, m', b, b') + D_r(A, A', L, L', r, r') + D_s(A, A', s, s')$$

The binding distance $D_b$, routing distance $D_r$, and scheduling distance $D_s$ now merely sum up values of the functions $d_b$, $d_r$, and $d_s$, respectively, for all possible options of both system synthesis problems, depending on both implementations. To obtain the relative distance from zero (implementations are identical) to one (implementation are completely different), we can normalize the four absolute distances by dividing by $|M_m \cup M_{m'}|$, $|L_{A,L} \cup L_{A',L'}|$, $|T_A \cup T_{A'}|$, and $|M_m \cup M_{m'}| + |L_{A,L} \cup L_{A',L'}| + |T_A \cup T_{A'}|$, respectively.

Returning to our child implementation in Figure 3, we can now select from among the 71,818 implementations using the distance measures.

Figure 4 shows an implementation that is very similar to the implementation in Figure 2. In fact, if we let $(A, P)$ be the system synthesis problem in Figure 1, $(A', P')$ the system synthesis problem in Figure 3, $(b, r, s)$ the implementation in Figure 2, and $(b', r', s')$ the implementation in Figure 4, then we have $D_b(m, m', b, b') = 1$, $D_r(A, A', L, L', r, r') = 6$, and $D_s(A, A', s, s') = 8$. Here, the binding and routing distance are minimal because distance one and six, respectively, are the lower bound of changes induced by removing task $t_1$. On the contrary, the scheduling distance has a maximum value of eight, as there are eight entities that have to be scheduled overall: four tasks and four dependencies, which are either removed or have different starting times.

This shows that the scheduling distance as defined above might be too granular for an effective comparison; we also observed this in our experiments, which we present in Section 6. Different scheduling distances could be employed, such as absolute distance between two start points, but this is computationally difficult, and we relegate this task to future work. Just considering the minimal binding and routing distance, we actually obtain an non-dominated child implementation in Figure 3.

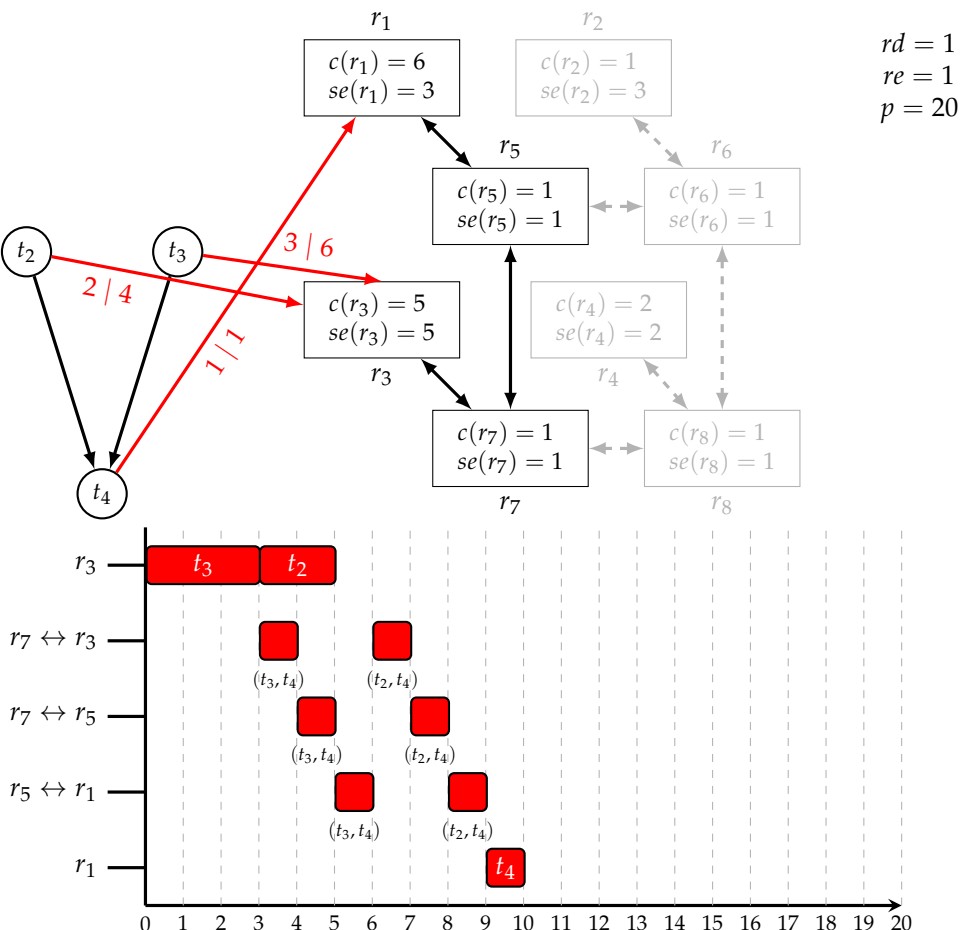

**Figure 4.** Example implementation (child implementation).

## 4. Encoding the System Synthesis Problem with ASP Modulo Difference Constraints

As is typical for ASP, our approach to solving the system synthesis problem with ASP modulo difference constraints (AMT) is also separated into a problem instance specifying the system synthesis problem and a general problem encoding. We start by describing the fact format; then, we present the general problem encoding facilitating binding, routing, and scheduling. We also describe how our encoding produces valid bindings, routings, and schedulings. This does not constitute a formal proof but rather intuitive explanations of the encoding techniques involved. We further provide different routing schema that use shortest path information to achieve better solution quality and runtime performance. Following that, we describe how multi-objective optimization is accomplished. Finally, we finish this section by providing the means of achieving evolutionary design space exploration, i.e., how we implement a similarity measure in ASP and how we can encourage finding similar solutions via strategies, preferences, and domain-specific heuristics.

### 4.1. Fact Format

Given a system synthesis problem $(A, ((R, L), rd, c, se, re, p, m, e, de))$, we create the following facts:

- `task(`$t$`)`, `send(`$t'$`,`$t''$`)` for $t \in T$, $(t', t'') \in D$ and $(T, D) \in A$;
- `link(`$r$`,`$r'$`)` for $(r, r') \in L$;
- `cost(`$r$`,`$c(r)$`)` for $r \in R$;
- `static_energy(`$r$`,`$se(r)$`)` for $r \in R$;
- `mapping(`$r$`,`$t$`)`, `execution(`$r$`,`$t$`,`$e(r,t)$`)`; and `dynamic_energy(`$r$`,`$t$`,`$de(r,t)$`)` for $t \in m(r)$ and $r \in R$;
- `routing_delay(`$rd$`)`, `routing_energy(`$re$`)`, and `period(`$p$`)`

For instance, Listing 1 shows the facts representing the system synthesis problem in Figure 1. We obtain the instance capturing the child implementation in Figure 3 by removing facts referencing `t1` from Listing 1 and changing values for `r1` to `cost(r1,6)`, `static_energy(r1,3)`, and `dynamic_energy(r1,t4,1)`.

Note that it is necessary to capture the existence of tasks via an atom, while this is not necessary for resources. This enables us to have single-task applications, and the resources can be extracted from either cost or static energy consumption.

### 4.2. General Problem Encoding

Listing 2 shows the succinct binding encoding that consists of one choice rule assigning exactly one resource to each task on which it can be mapped. Atoms over predicate `bind/2` describe the binding $b$. That is, `bind(`$t$`,`$r$`)` represents that $b(t) = r$ for task $t$ and resource $r$. Bindings can only be chosen among possible mappings and are therefore valid. For instance, facts `bind(t1,r1)`, `bind(t2,r3)`, `bind(t3,r3)`, and `bind(t4,r1)` capture the binding of the parent implementation in Figure 2.

In Listing 3, we find the routing encoding. The routing $r$ is captured via atoms over predicate `route/3`. In more detail, for every atom `route((`$t$`,`$t'$`),`$u$`,`$u'$`)`, we have $(u, u') \in r((t, t'))$ for dependency $(t, t')$ and link $(u, u')$. Choice rules in Lines 3 and 4 allow every resource in the network at most one outgoing route and at most one ingoing route per communication, respectively. As a result, every communication now has possibly disconnected paths or cyclic routes; however, we ensure that there is no branching. That is, at this point, routes are possibly disconnected or are cycles. Lines 6 to 10 enforce that for each communication, routes are connected and start and end at the correct resources. First, we require that routes are connected by recursively deriving resources that are visited via routes connected to the resource with the sending task. Then, we state that no route may end at a resource that is not visited, i.e., not connected to the sender of the communication. Second, we select the resource with the receiving task and require this resource to be visited and disallow routes originating from there. This expresses that the communication arrives at the target location, and we stop once this is achieved.

**Listing 1.** Facts representing our example parent specification in Figure 1.

```
1  task(t1). task(t2). task(t3). task(t4).
2  send(t1,t2). send(t1,t3). send(t2,t4). send(t3,t4).

4  link(r1,r5). link(r5,r1). link(r2,r6). link(r6,r2).
5  link(r3,r7). link(r7,r3). link(r4,r8). link(r8,r4).
6  link(r5,r6). link(r6,r5). link(r5,r7). link(r7,r5).
7  link(r6,r8). link(r8,r6). link(r7,r8). link(r8,r7).

9  cost(r1,3). cost(r2,1).
10 cost(r3,5). cost(r4,2).
11 cost(r5,1). cost(r6,1).
12 cost(r7,1). cost(r8,1).

14 static_energy(r1,5). static_energy(r2,3).
15 static_energy(r3,5). static_energy(r4,2).
16 static_energy(r5,1). static_energy(r6,1).
17 static_energy(r7,1). static_energy(r8,1).

19 mapping(r1,t1). mapping(r1,t4).
20 mapping(r2,t2).
21 mapping(r3,t2). mapping(r3,t3).
22 mapping(r4,t1). mapping(r4,t4).

24 execution(r1,t1,2). execution(r1,t4,1).
25 execution(r2,t2,2).
26 execution(r3,t2,2). execution(r3,t3,3).
27 execution(r4,t1,3). execution(r4,t4,1).

29 dynamic_energy(r1,t1,5). dynamic_energy(r1,t4,4).
30 dynamic_energy(r2,t2,3).
31 dynamic_energy(r3,t2,4). dynamic_energy(r3,t3,6).
32 dynamic_energy(r4,t1,3). dynamic_energy(r4,t4,1).

34 routing_delay(1).
35 routing_energy(1).
36 period(20).
```

**Listing 2.** Binding encoding.

```
1  1 { bind(T,R) : mapping(R,T) } 1 :- task(T).
```

**Listing 3.** Routing encoding.

```
1  resource(R;R') :- link(R,R').

3  { route((T,T'),R,R') : link(R,R') } 1 :- resource(R), send(T,T').
4  { route((T,T'),R,R') : link(R,R') } 1 :- resource(R'), send(T,T').

6  visit((T,T'),R) :- send(T,T'), bind(T,R).
7  visit(C,R')     :- visit(C,R), route(C,R,R').
8  :- route(C,_,R), not visit(C,R).
9  :- send(T,T'), bind(T',R), not visit((T,T'),R).
10 :- send(T,T'), bind(T',R), route((T,T'),R,_).
```

Because we may only choose our routing among links, Condition 1 is trivially satisfied. Condition 2 is met as we may not start a route from the target resource of a communication, and if that is the same as the source resource, we have no route. Finally, Condition 3 is satisfied, firstly because routing may only take place over visited resources, and the only resource that is visited without condition is the resource the sending task is bound to;

secondly, the route ends at the receiving task's resource because it has to be visited, but no further routing from that point is allowed. Therefore, routings produced by this encoding are valid. Note that this encoding does not produce all valid routings but only valid routings with acyclic routes. To facilitate cyclic routes, one must add an additional order when resources are visited, akin to a time step in action encodings. We decided against this since cycles offer little merit because we use a circuit switching scheduling strategy.

For our parent implementation in Figure 2, the atoms capturing the routing are as follows:

```
route((t1,t2),r1,r5)    route((t2,t4),r3,r7)
route((t1,t2),r5,r7)    route((t2,t4),r7,r5)
route((t1,t2),r7,r3)    route((t2,t4),r5,r1)

route((t1,t3),r1,r5)    route((t3,t4),r3,r7)
route((t1,t3),r5,r7)    route((t3,t4),r7,r5)
route((t1,t3),r7,r3)    route((t3,t4),r5,r1)
```

For several conditions on valid schedulings, we require the number of links used in every communication, the so-called *hops*. Listing 4 achieves this goal via a technique called chaining. First, we need to find an upper bound to the possible number of hops so that grounding is possible. This is the number of links in the network (Line 1). Then, we initialize the count with zero at the resource where the sending task is bound to (Line 2). Recursively, we add one to our count for every link the communication uses in Line 3 if we have not yet reached the maximum limit. Finally, we can determine the number of hops by selecting the hop count at the resource that the receiving task is bound to (Line 5). In detail, atom hops($d$,$n$) determines that communication $d$ takes $n$ hops. Note that we can use the #count directive to count the links as this is static information and the count is determined at grounding time. On the other hand, it would lead to an exponential number of rules if we encoded the counting of the hops in the same way because all possible subsets of routes taken need to be considered. Our chaining encoding scales with the number of links multiplied by the number of resources. So, in the worst case, it is cubic in the number of resources and has better propagation properties compared with the #count-based method.

**Listing 4.** Calculating number of hops for every communication.

```
1  nr_links(NR)   :- NR=#count{ link(N,N') : link(N,N') }.
2  hops((T,T'),R,0)     :- send(T,T'), bind(T,R).
3  hops(C,R',H+1)       :- hops(C,R,H), route(C,R,R'),
4  H<NR, nr_links(NR).
5  hops(C,H)            :- hops(C,R,H), send(T,T'), bind(T',R).
```

Because scheduling relies on deciding what tasks to execute first on the same resource, or what communications to send first if they use the same link, we need to facilitate that choice in our encoding. Listing 5 detects conflicts among tasks and communications and assigns priorities via atoms over predicate priority/2. In detail, atom priority($t$,$t'$) expresses that $t$ is executed before $t'$ may start for $t \in \mathbb{T} \cup \mathbb{D}$. In order to narrow the search for conflicts, we first analyze which tasks and communications depend on each other, i.e., which are sequentially executed in one application and therefore cannot be simultaneously executed. We first determine direct dependencies in Lines 1 to 2; here, task dependencies are directly given and communication dependencies arise whenever the receiver task of a communication becomes a sender task of another communication. Then, we build the transitive closure in Lines 4 to 5. Note that this is usually expensive with respect to memory and performance, but here, we only build the transitive closure on domain predicates that are decided during grounding. The rule in Line 7 determines the conflicts between resources. Two tasks are in a conflict if they are bound to the same resource and they do not depend on each other. Note that the grounding is quadratic in the number of tasks. We reduce the problem size by ordering the tasks alphanumerically and by only considering

conflicts where the first task is smaller. We explain in the scheduling encoding that this does not lead to a loss of information. Similarly, Line 12 determines conflicts between communications. Two communications are in conflict whenever they share at least one link and do not depend on each other. We use the same alphanumeric ordering as for resource conflicts. Finally, Line 19 decides the priority between two tasks or communications that are in conflict. Again, the choice rule only allows for deciding that an alphanumerically smaller task or communication is executed first, but we take the absence of this priority to mean the opposite.

**Listing 5.** Conflict detection and resolution.

```
1  depends(T,T')  :- send(T,T').
2  depends((T,T'),(T',T''))  :- send(T,T'), send(T',T'').

4  depends_trans(T,T') :- depends(T,T').
5  depends_trans(T,T') :- depends_trans(T,T'), depends(T,T').

7  conflict(T,T') :- task(T), task(T'), T < T',
8  bind(T,R), bind(T',R),
9  not depends_trans(T',T),
10 not depends_trans(T,T').

12 conflict((T,T'),(T'',T''')) :- send(T,T'), send(T'',T'''),
13 (T,T') < (T'',T'''),
14 1 #sum{ 1 : route((T,T'),R,R'),
15 route((T'',T'''),R,R')},
16 not depends_trans((T,T'),(T'',T''')),
17 not depends_trans((T'',T'''),(T,T')).

19 {priority(C,C')} :- conflict(C,C').
```

Listing 6 gives our scheduling encoding. We facilitate scheduling via difference constraints, where the name of a task or communication is the name of the integer variable that represents their starting time in scheduling $s$. In the following, we argue on the ground level and with the mathematical entities of the system synthesis problem. That is, every `T` and `T'` in the encoding is instantiated by all tasks $\{t, t'\} \subseteq \mathbb{T}$, as well as every `C` and `C'` is instantiated with all communications $\{c, c'\} \subseteq \mathbb{D}$. Lines 1 to 2 implement Condition 1 on a valid scheduling; they derive difference constraint `&diff{0-t}<=0` ensuring $0 \leq s(t)$ and `&diff{t-0}<=p` $- e(b(t), t)$ ensuring $s(t) + e(b(t), t) \leq p$ for $t \in \mathbb{T}$. Lines 4 and 5 handle dependencies between tasks and communications. Specifically, Line 4 encodes Condition 2 by deriving difference constraint `&diff{t-(t,t')}<=`$-e(b(t), t)$ for every dependency, which means that $s(t) + e(b(t), t) \leq s((t, t'))$ for $(t, t') \in \mathbb{D}$. Similarly, Line 5 handles Condition 3 via difference constraint `&diff{(t,t')-t'}<=`$-|r((t, t'))| * rd$, where the number of hops is used to determine $|r((t, t'))|$. Then, we have $s((t, t')) + |r((t, t'))| * rd \leq s(t')$ for $(t, t') \in \mathbb{D}$. Lines 8 to 13 handle conflict resolution between tasks. For two conflicting tasks $\{t, t'\} \subseteq \mathbb{T}$ with $t \neq t'$ so that $b(t) = b(t')$, we obtain either difference constraint `&diff{t-t'}<=`$-e(b(t), t)$ enforcing $s(t) + e(b(t), t) \leq s(t')$ if $t$ is prioritized over $t'$ or difference constraint `&diff{t'-t}<=`$-e(b(t'), t')$ and therefore $s(t') + e(b(t'), t') \leq s(t)$ if there is no prioritization. Note that the conflict relation is symmetrical; thus, we only need to consider one ordering of the task names to obtain Condition 4. A drawback when compared wtih symmetrical conflict and priority atoms is having to duplicate the rules that derives the difference constraints in the encoding; however, the number of rule instantiations overall is halved. Similarly, Lines 15 to 18 resolve communication conflicts. For two conflicting communications $\{d, d'\} \subseteq \mathbb{D}$ with $d \neq d'$, i.e., we have $r(d) = (\ldots, r, r', \ldots)$ and $r(d') = (\ldots, r, r', \ldots)$, we derive either difference constraint `&diff{d-d'}<=`$-|r(d)| * rd$ leading to $s(d) + |r(d)| * rd \leq s(d')$ if $d$ has priority over $d'$ or difference constraint `&diff{d'-d}<=`$-|r(d')| * rd$ giving rise to $s(d') + |r(d')| * rd \leq s(d)$ if

no priority exists. Again, Condition 5 is satisfied by only considering one ordering of the communications due to symmetry, and the symmetry breaking technique is identical to the one used for resource conflicts. Now, our encoding produces valid schedules as we implemented all conditions via difference constraints.

**Listing 6.** Scheduling encoding.

```
1  &diff { 0-T } <= 0 :- task(T).
2  &diff { T-0 } <= V :- period(P), bind(T,R), execution(R,T,E), V=P-E.

4  &diff { T-(T,T') } <= -E :- send(T,T'), bind(T,R), execution(R,T,E).
5  &diff { (T,T')-T'} <= -S :- send(T,T'), hops(C,N), routing_delay(D),
6  S=N*D.

8  &diff { T-T'} <= -E :- conflict(T,T'), priority(T,T'),
9  task(T), task(T'),
10 execution(R,T,E).
11 &diff { T'-T} <= -E :- conflict(T,T'), not priority(T,T'),
12 task(T), task(T'),
13 execution(R,T',E).

15 &diff { C-C' } <= -S :- conflict(C,C'), priority(C,C'),
16 hops(C,N), routing_delay(D), S=N*D.
17 &diff { C'-C } <= -S :- conflict(C,C'), not priority(C,C'),
18 hops(C',N), routing_delay(D), S=N*D.
```

For example, the valid schedule of our parent implementation in Figure 2 is

```
dl(t1,0)        dl(t2,8)        dl(t3,5)        dl(t4,14)
dl((t1,t2),5) dl((t1,t3),2)   dl((t2,t4),11) dl((t3,t4),8)
```

### 4.2.1. Routing Variants

In this section, we show how we can use information about the location of resources to obtain shortest path information and improve solution quality and runtime performance. More specifically, we view our architecture as a three-dimensional grid on which our resources have specific locations that can be used to easily calculate the shortest paths between them. For that purpose, we define a function $l : R \rightarrow \mathbb{N} \times \mathbb{N} \times \mathbb{N}$ that assigns each resource their location and, given a set of resources $R$, add facts `location(r,l(r))` to our instance for each resource $r \in R$. We use this information to firstly restrict routes to the length of the shortest path possible to improve solution quality and solving time; secondly, we use this information to enable dimension-ordered routing, which essentially removes routing from the solving process, i.e., the possible routes are known at grounding time.

Listing 7 shows the coordinates of resources of our parent specification in Figure 1. Note that the computational resources have the same coordinates as the routers they are connected to. The scheme we use for our instances assumes that every computational resource has a router that it is closely and exclusively attached to, and the shortest paths are then considered between the routers. In that sense, we have a two-dimensional NoC in our example as the z-coordinate is always zero.

**Listing 7.** Locations of resources of the parent specification.

```
1  location(r1,(0,1,0)). location(r2,(1,1,0)).
2  location(r3,(0,0,0)). location(r4,(1,0,0)).
3  location(r5,(0,1,0)). location(r6,(1,1,0)).
4  location(r7,(0,0,0)). location(r8,(1,0,0)).
```

Bound Routing

Listing 8 shows how we calculate the number of hops and restrict the routes using the locations of resources. Line 1 calculates the former for two tasks that are bound to different resources. The number of hops required to route the message is exactly the Manhatten distance between the coordinates plus two. The two hops are added to account for the segment from the sending resource to its router and from the final router to the receiving resource. Line 4 covers the case when both tasks in communication are bound to the same resource, where the number of hops is then zero. Finally, Line 5 restricts the routing of every communication to the calculated number of hops, where it is either the shortest path across the network or zero depending on the binding.

**Listing 8.** Bind number of hops to shortest path according to grid coordinates.

```
1  hops((T,T'),N) :- bind(T,R), bind(T',R'), R!=R', send(T,T'),
2  location(R,(X,Y,Z)), location(R',(X',Y',Z')),
3  N = |X-X'|+|Y-Y'|+|Z-Z'|+2.
4  hops((T,T'),0) :- bind(T,R), bind(T',R), send(T,T').
5  :- hops(C,N), not N { route(C,_,_) : route(C,_,_) } N.
```

Our parent implementation in Figure 2 fulfills this property. We have as coordinates $(0,1,0)$ and $(0,0,0)$ for `r1` and `r3`, respectively. Then, the shortest number of hops is $|0-0|+|1-0|+|0-0|+2 = 3$ for all communications, which is the amount each communication requires in this implementation. This encoding ensures the shortest paths while allowing for variable routes. Furthermore, we restrict the search space, which should lead to an improvement in the solving performance.

Dimension-Ordered Routing

Listing 9 shows the encoding for dimension-ordered routing. The basic idea is to find a path from the source to the target that first diminishes the distance in the x-coordinates until no distance remains, then the same for the y-coordinates and z-coordinates accordingly. For that purpose, the encoding creates a routing table that is stored in atoms over predicate `next/3` (Lines 1–13). Essentially, an atom `next((x,y,z),(x',y',z'),(x'',y'',z''))` indicates that the shortest path from $(x,y,z)$ to $(x',y',z')$ involves the link from $(x,y,z)$ to $(x'',y'',z'')$. For every two locations $(x,y,z)$ and $(x',y',z')$, the rules identify in order whether x-, y-, or z-coordinates are equal and make the case distinction of whether one has to increment or decrement the respective unequal coordinate to get closer to the target. Depending on the case, x-, y-, or z-coordinate are incremented or decremented by one to determine the next destination. Then, Lines 14 to 21 use the routing table to determine the route for all communications given the specific binding of the tasks involved. Furthermore, we can use the same calculation in Listing 8 to determine the number of hops. Note that routing now completely depends on binding and no longer involves independent planning. As above, we guarantee the shortest paths and we restrict the search space even further. However, this routing is the most inflexible, so we might discard implementations that achieve a better quality as communication may evade each other with the more flexible routing variants.

4.2.2. Preference Encoding

This section describes how we implement Pareto optimization. To describe preferences, we use the fact format of the system *asprin* described in [13], and to achieve grounding, solving, and optimization, we employ a Python script based on *clingo*'s theory and application framework [14]; the script is a re-implementation of the design space exploration system described in [2]. The general idea is that difference constraints as well as Pareto optimization are handled by background theories that are integrated via *clingo*'s theory interface. While the former sanctions thne validity of answer sets with respect to schedulability, the latter holds an archive of currently non-dominated implementations, updates this archive when new and better solutions are found, and adds conflict clauses whenever (partial) answer

sets are dominated by members of the archive. In this way, the quality of implementations in the archive improves over time as more answer sets are found; eventually, the archive is equal to the Pareto front once no more non-dominated implementations can be found.

**Listing 9.** Routing encoding via dimension-ordered routing.

```
1  coord(X,Y,Z) :- location(_,(X,Y,Z)).
2  next((X,Y,Z),(X',Y',Z'),(X+1,Y,Z)) :- coord(X,Y,Z), coord(X',Y',Z'),
3  coord(X+1,Y,Z), X < X'.
4  next((X,Y,Z),(X',Y',Z'),(X-1,Y,Z)) :- coord(X,Y,Z), coord(X',Y',Z'),
5  coord(X-1,Y,Z), X > X'.
6  next((X,Y,Z),(X,Y',Z'),(X,Y+1,Z))  :- coord(X,Y,Z), coord(X,Y',Z'),
7  coord(X,Y+1,Z), Y < Y'.
8  next((X,Y,Z),(X,Y',Z'),(X,Y-1,Z))  :- coord(X,Y,Z), coord(X,Y',Z'),
9  coord(X,Y-1,Z), Y > Y'.
10 next((X,Y,Z),(X,Y,Z'),(X,Y,Z+1))   :- coord(X,Y,Z), coord(X,Y,Z'),
11 coord(X,Y,Z+1), Z < Z'.
12 next((X,Y,Z),(X,Y,Z'),(X,Y,Z-1))   :- coord(X,Y,Z), coord(X,Y,Z'),
13 coord(X,Y,Z-1), Z > Z'.
14 route((T,T'),R,R')  :- bind(T,R), bind(T',R'), send(T,T'),
15 location(R,C), location(R',C), R!=R'.
16 route((T,T'),R,R'') :- route((T,T'),_,R), bind(T',R'),
17 location(R,C), location(R',C'),
18 next(C,C',C''), location(R'',C''),
19 not bind(T',R'').
20 route((T,T'),R',R)  :- bind(T',R), send(T,T'),
21 location(R,C), location(R',C), R!=R'.
```

To determine the quality of an implementation, we first determine which resources are allocated. This can be seen in Listing 10. Here, we derive atoms over predicate `allocated/1`, where `allocated(r)` signifies that resource $r$ is used in an implementation. A resource is considered to be in use if a task is bound to it (Line 1) or if it is included in the routing of a communication (Lines 3–4).

**Listing 10.** Deriving allocation of an implementation.

```
1  allocated(R) :- bind(T,R).

3  allocated(R) :- route(_,R,_).
4  allocated(R) :- route(_,_,R).
```

The preference encoding in Listing 11 makes use of this information. Preference definitions rely on atoms over predicates `preference/2` and `preference/5`. Additionally, rules are provided that capture atoms that the preferences relate to via atoms over predicate `holds/2`. These predicates are known to the Pareto optimization system and are used to communicate the desired preferences. All preference definitions follow the same scheme. First, an atom `preference(n,t)` declares preference $n$ of type $t$; in our case, we have `preference(cost,sum)`, `preference(energy,sum)`, and `preference(latency,max)`. Here, `sum` and `max` are predefined preference types that are known to our Pareto preference propagator. The former determines the sum of elements that are added if certain conditions hold, and the latter determines the maximum of integer variables, optionally with a constant offset. For those preference types, smaller values are preferred, and we use Pareto optimization over all preferences. In contrast to *asprin*, preference types cannot be added via ASP but are Python classes that implement certain methods. This is due to the fact that the values of the start times have to be considered to calculate the latency. These values are only known to the difference constraints propagator and are never reified in the ASP encoding. Then, the preference elements are defined, i.e., what quality holds under

what conditions. To determine the cost of an implementation, the rule in Line 2 states that whenever `allocated(r)` holds, we add $c(r)$ to the value of preference `cost` for $r \in R$. The energy consumption is split into several preference statements. Line 7 adds $se(r)$ whenever `allocated(r)` holds for $r \in R$. Line 11 sums up the energy consumption of every link used in a communication by adding $re$ for every `route((t,t'),r,r')` that holds for $(t,t') \in \mathbb{D}$ and $\{r,r'\} \subseteq R$. Finally, Line 15 states that whenever `bind(t,r)` holds, we add $se(r,t)$ for $r \in R$ and $t \in \mathbb{T}$. For latency, Line 21 states that whenever `bind(t,r)` holds, we add the value $s(t) + e(r,t)$ to the set over which the maximum is calculated for $t \in \mathbb{T}$ and $r \in R$. Essentially, we build the maximum over all starting times plus execution times, which is the latency of the implementation. The third and final part of each preference description reifies atoms that are conditions for the preference elements into `holds/1` atoms. Note that this is the only non-domain part of the preference encoding. In detail, `holds/1` atoms describe the part of the answer set that determines the quality. For more information on the fact format and methodology see [2,13]. Similar to *clingo*[DL], our Pareto optimization framework adds symbols to the answer set and describes the quality of the solution. Specifically, we add symbols of the form `pref(n,t,v)`, where $n$ is the name, $t$ is the type, and $v$ is the objective value.

**Listing 11.** Encoding preferences cost, energy consumption, and latency.

```
1   preference(cost,sum).
2   preference(cost,(1,1),1,for(atom(allocated(R))),(C))
3   :- cost(R,C).
4   holds(atom(allocated(R)),0) :- allocated(R).

6   preference(energy,sum).
7   preference(energy,(2,1),1,for(atom(allocated(R))),(S))
8   :- static_energy(R,S).
9   holds(atom(allocated(R)),0)
10  :- allocated(R).
11  preference(energy,(2,2),1,for(atom(route((T,T'),R,R'))),(E))
12  :- link(R,R'), send(T,T'), routing_energy(E).
13  holds(atom(route((T,T'),R,R')),0)
14  :- route((T,T'),R,R').
15  preference(energy,(2,3),1,for(atom(bind(T,R))),(D))
16  :- mapping(R,T), dynamic_energy(R,T,D).
17  holds(atom(bind(T,R)),0)
18  :- bind(T,R).

20  preference(latency,max).
21  preference(latency,(3,1),1,for(atom(bind(T,R))),(T,E))
22  :- mapping(R,T), execution(R,T,E).
23  holds(atom(bind(T,R)),0)
24  :- bind(T,R).
```

For instance, our parent implementation in Figure 2 has quality `pref(cost,sum,10)`, `pref(energy,sum,43)`, and `pref(latency,max,15)`.

## 5. Encoding Evolutionary Design Space Exploration

In this section, we describe how we implement an evolutionary design space exploration with our ASP-based framework. We start by outlining how we encode the similarity measure between a parent and child implementation. For that, we only focus on the binding and routing distance as the scheduling similarity performed badly in our empirical analysis (cf. Section 6). Then, we present three techniques for facilitating similar child implementations: strategies, preferences, and domain-specific heuristics. In essence, *strategies* restrict the search space via integrity constraints to only contain similar implementations, and *preferences* use the distance measures as another entry in the quality and therefore make

it subject to Pareto optimization; *domain-specific heuristics* adapt the solver's heuristics in such a way that similar implementations are emphasized. Note that only strategies remove solutions while preferences and heuristics consider the whole search space. For all of these techniques, we consider both encouraging similarity and discouraging dissimilarity.

*5.1. Encoding the Similarity Measure*

To establish the similarity measure between parent and child, we first reify all atoms of the parent specification and parent implementation in atoms over predicated `parent/1`. For instance, the atom `link(r1,r5)` is a member of the parent specification in Figure 1, and `route((t1,t2),r1,r5)` is contained in the parent implementation in Figure 2; so, we add facts `parent(link(r1,r5))` and `parent(route((t1,t2),r1,r5))`, respectively. Listing 12 shows the facts that are relevant to establish similarity between parent and child in our running example. Listing 13 shows the encoding of the similarity measure; it derives two kinds of atoms: `equal/1` atoms that indicate that an aspect of the parent and child implementation are equal and `unequal/1` atoms capturing a difference between the parent and child implementation. The encodings closely follow the cases of the definition of functions $d_b$ and $d_r$. Lines 1 to 5 establish equal bindings of the parent and child, i.e., the "otherwise" case of $d_b$. Parent and child implementation are equal for the binding of task $t$ on resource $r$ if they either both bound $t$ to $r$ or the mapping exists in either the child or parent specification but both did not use this mapping. For instance, we derive `equal(bind(t4,r1))` and `equal(bind(t1,r4))` for our running example; the first is derived because both the parent and child implementation have the same binding, and the second is derived because the binding exists in the parent specification but both implementations do not use it. The encoding of unequal bindings in Lines 6 to 13 follows the remaining cases of function $d_b$. Parent and child implementation are unequal for a binding of task $t$ on resource $r$ if a mapping is only possible in either the child or parent specification and the respective binding is part of the respective implementation or the mapping is possible in both, but is unequal in the parent and child implementations. Returning to the examples, we would have `unequal(bind(t1,r1))` because `t1` no longer exists in the child specification. Other than that, the binding of the parent and child implementation in Figure 2 and 4 are identical. Establishing similarity with respect to the routing follows a very similar pattern. Lines 15 to 22 encode the equality of the routing, again corresponding to the "otherwise" case of function $d_r$. Parent and child implementation are equal for a link $(r, r')$ and communication $d$ if either both implementations use link $(r, r')$ for routing communication $d$ or neither uses it but the possibility exists in either the parent or child specification; in our example, we have `equal(route((t3,t5),r3,r7))` because both parent and child implementation use this route and `equal(route((t1,t2),r5,r6))` because neither implementation contains it, but it is possible in the parent specification. The similarity encoding for the unequal parts of the parent and child implementation follows the cases of function $d_r$. Parent and child implementation are unequal for a link $(u, u')$ and communication $d$ if $d$ is part of only either child or parent implementation and $(u, u')$ belongs to the routing of $d$ in the respective implementation or if $d$ is part of both specifications, but $(u, u')$ is part of the routing of $d$ in either the child or parent implementation only. Note that we do not need to specifically refer to all possible links in the rules because `route/3` atoms are grounded over all possible links, and only the existence of a route in either parent or child implementation can lead to a dissimilarity. Let us return to our example. All routes that involve communications with task `t1` are unequal from parent to child implementation. Specifically, we derive:

```
unequal(route((t1,t2),r1,r5))  unequal(route((t1,t2),r5,r7))
unequal(route((t1,t2),r7,r3))  unequal(route((t1,t3),r1,r5))
unequal(route((t1,t3),r5,r7))  unequal(route((t1,t3),r7,r3))
```

All other aspects of routing are identical. Note that we obtain the value for functions $D_b$ and $D_r$ if we count the `unequal/1` atoms for binding and routing, respectively. For our example, we have one unequal atom for binding and six for routing, which are the values calculated by the distance functions. Note that we could derive `equal/1` atoms

from unequal/1 atoms and vice-versa but decided to give an independent definition to make all cases explicit. Furthermore, we can see that equal/1 atoms are derived for all possibilities in both specifications, not only for bindings and routings that are used in either the child or parent implementation. On the other hand, unequal/1 atoms are possible for bindings and routings that are part of at least one implementation. We therefore suspect different behaviors when using one or the other. In the following strategies, preferences, and heuristics, we separate the use of both kinds of atoms, and in Section 6, we empirically analyze the different impact.

*5.2. Strategies*

Listings 14 and 15 show how we use strategies that forbid unequal and enforce equal implementations with respect to binding and routing, respectively. The integrity constraints in Listing 14 simply forbid that any binding or routing is unequal. On the other hand, Listing 15 forbids that any possible mapping or routing is not equal. Note that we have to use negation in Listing 15; therefore, we need to provide the domain of the equal/1 atoms, which are the possible mappings and routings in both the parent and child specifications. These strategies are our most invasive way to foster similar implementations. On the one hand, we expect that this leads to unsatisfiability or an exclusion of some high-quality implementations; on the other hand, in the case of satisfiability, we can ensure similarity and achieve a good runtime performance. For instance, our running example is unsatisfiable with these strategies as the removal of a task implies some inequality in binding and routing.

**Listing 12.** Relevant parts of parent specification and implementation to establish similarity.

```
1   parent(send(t1,t2)).  parent(send(t1,t3)).
2   parent(send(t2,t4)).  parent(send(t3,t4)).

4   parent(link(r1,r5)).  parent(link(r5,r1)).
5   parent(link(r2,r6)).  parent(link(r6,r2)).
6   parent(link(r3,r7)).  parent(link(r7,r3)).
7   parent(link(r4,r8)).  parent(link(r8,r4)).
8   parent(link(r5,r6)).  parent(link(r6,r5)).
9   parent(link(r5,r7)).  parent(link(r7,r5)).
10  parent(link(r6,r8)).  parent(link(r8,r6)).
11  parent(link(r7,r8)).  parent(link(r8,r7)).

13  parent(mapping(r1,t1)).  parent(mapping(r1,t4)).
14  parent(mapping(r2,t2)).
15  parent(mapping(r3,t2)).  parent(mapping(r3,t3)).
16  parent(mapping(r4,t1)).  parent(mapping(r4,t4)).

18  parent(bind(t1,r1)).
19  parent(bind(t2,r3)).
20  parent(bind(t3,r3)).
21  parent(bind(t4,r1)).
22  parent(route((t1,t2),r1,r5)).
23  parent(route((t1,t2),r5,r7)).
24  parent(route((t1,t2),r7,r3)).
25  parent(route((t1,t3),r1,r5)).
26  parent(route((t1,t3),r5,r7)).
27  parent(route((t1,t3),r7,r3)).
28  parent(route((t2,t4),r3,r7)).
29  parent(route((t2,t4),r5,r1)).
30  parent(route((t2,t4),r7,r5)).
31  parent(route((t3,t4),r3,r7)).
32  parent(route((t3,t4),r5,r1)).
33  parent(route((t3,t4),r7,r5)).
```

**Listing 13.** Deriving `equal/1` and `unequal/1` atoms to capture similarity between implementations.

```
1  equal(bind(T,R)) :- bind(T,R), parent(bind(T,R)).
2  equal(bind(T,R)) :- not bind(T,R), not parent(bind(T,R)),
3  mapping(R,T).
4  equal(bind(T,R)) :- not bind(T,R), not parent(bind(T,R)),
5  parent(mapping(R,T)).
6  unequal(bind(T,R)) :- bind(T,R), mapping(R,T),
7  not parent(mapping(R,T)).
8  unequal(bind(T,R)) :- parent(bind(T,R)), not mapping(R,T),
9  parent(mapping(R,T)).
10 unequal(bind(T,R)) :- bind(T,R), not parent(bind(T,R)),
11 mapping(R,T'), parent(mapping(R,T')).
12 unequal(bind(T,R)) :- not bind(T,R), parent(bind(T,R)),
13 mapping(R,T'), parent(mapping(R,T')).

15 equal(route((T,T'),R,R')) :- route((T,T'),R,R'),
16 parent(route((T,T'),R,R')).
17 equal(route((T,T'),R,R')) :- not route((T,T'),R,R'),
18 not parent(route((T,T'),R,R')),
19 send(T,T'), link(R,R').
20 equal(route((T,T'),R,R')) :- not route((T,T'),R,R'),
21 not parent(route((T,T'),R,R')),
22 parent(send(T,T')), parent(link(R,R')).
23 unequal(route((T,T'),R,R')) :- route((T,T'),R,R'),
24 not parent(send(T,T')), send(T,T').
25 unequal(route((T,T'),R,R')) :- parent(route((T,T'),R,R')),
26 parent(send(T,T')),  not send(T,T').
27 unequal(route((T,T'),R,R')) :- route((T,T'),R,R'),
28 not parent(route((T,T'),R,R')),
29 send(T,T'), parent(send(T,T')).
30 unequal(route((T,T'),R,R')) :- not route((T,T'),R,R'),
31 parent(route((T,T'),R,R')),
32 send(T,T'), parent(send(T,T')).
```

**Listing 14.** Strategy not allowing `unequal/1` atoms.

```
1  :- unequal(bind(T,R)).
2  :- unequal(route((T,T'),R,R')).
```

**Listing 15.** Strategy enforcing `equal/1` atoms.

```
1  :- not equal(bind(T,R)), mapping(R,T).
2  :- not equal(bind(T,R)), parent(mapping(R,T)).

4  :- not equal(route((T,T'),R,R')), send(T,T'), link(R,R').
5  :- not equal(route((T,T'),R,R')), parent(send(T,T')), parent(link(R,R')).
```

### 5.3. Preferences

Listings 16 and 17 use the same fact format as described in Section 4.2.2 to add a fourth objective value: the distance of the child implementation to the parent implementation. While the former punishes unequal parts of the implementations, the latter encourages equality. In detail, we define a preference `dist`, which is subject to type `sum`. In Listing 16 Lines 3 to 8, we add preference elements that add one to the objective value whenever a possible binding is not equal between the child and parent implementations. Lines 10 to 15 define the same for all possible routings accordingly. Listing 17, on the other hand, adds minus one for every equal binding in Lines 3 to 8 and possible routing in Lines 10 to 15. Because the objectives are subject to minimization, adding negative values expresses that equality is preferred. Here, we include similarity information into the construction of

the Pareto front. Eventually, the Pareto front would include the closest possible solution with the best possible objective value, i.e., the solution with the highest quality that is as close as possible, and the non-dominated trade-offs in between. The child implementation in Figure 4 in comparison to the parent implementation in Figure 2 with preference encoding in Listing 16 results in objective value `pref(dist,sum,7)` and with Listing 17 in `pref(dist,sum,-101)`.

**Listing 16.** Preference discouraging `unequal/1` atoms.

```
1   preference(dist,sum).

3   preference(dist,(4,1),1,for(atom(unequal(bind(T,R)))), (1))
4   :- mapping(R,T).
5   preference(dist,(4,2),1,for(atom(unequal(bind(T,R)))), (1))
6   :- parent(mapping(R,T)).
7   holds(atom(unequal(bind(T,R))),0)
8   :- unequal(bind(T,R)).

10  preference(dist,(4,3),1,for(atom(unequal(route((T,T'),R,R')))), (1))
11  :- send(T,T'), link(R,R').
12  preference(dist,(4,4),1,for(atom(unequal(route((T,T'),R,R')))), (1))
13  :- parent(send(T,T')), parent(link(R,R')).
14  holds(atom(unequal(route((T,T'),R,R'))),0)
15  :- unequal(route((T,T'),R,R')).
```

**Listing 17.** Preference encouraging `equal/1` atoms.

```
1   preference(dist,sum).

3   preference(dist,(4,1),1,for(atom(equal(bind(T,R)))), (-1))
4   :- mapping(R,T).
5   preference(dist,(4,2),1,for(atom(equal(bind(T,R)))), (-1))
6   :- parent(mapping(R,T)).
7   holds(atom(equal(bind(T,R))),0)
8   :- equal(bind(T,R)).

10  preference(dist,(4,3),1,for(atom(equal(route((T,T'),R,R')))), (-1))
11  :- send(T,T'), link(R,R').
12  preference(dist,(4,4),1,for(atom(equal(route((T,T'),R,R')))), (-1))
13  :- parent(send(T,T')), parent(link(R,R')).
14  holds(atom(equal(route((T,T'),R,R'))),0)
15  :- equal(route((T,T'),R,R')).
```

### 5.4. Domain-Specific Heuristics

Listings 18 and 19 show how we modify the heuristics of `equal/1` and `unequal/1` atoms, respectively. Note that we use constants `value` and `modifier` instead of a heuristic mode or value for the first two lines. This allows us to modify the importance of the atoms in terms of the branching heuristics via the command line, i.e., we modify to which degree the solver emphasizes choices over atoms that express similarity; although, we always provide the sign heuristic for `equal/1` and `unequal/1` atoms. We modify `equal/1` atoms with a positive sign to encourage similarity and `unequal/1` atoms with a negative sign to avoid dissimilarity. The advantage of such heuristics is that the search space remains intact, they do not introduce additional complexity in the form of a new objective value, and they usually lead to good solutions fast. The drawback is that we cannot make any guarantees about similarity, and domain-specific heuristics usually hinder the optimization process. This is due to the fact that the optimization terminates with an unsatisfiability proof for which the domain-specific heuristics is at minimum useless or even actively derails the solving.

**Listing 18.** Heuristics discouraging `unequal/1` atoms.

```
1  #heuristic equal(bind(T,R)). [value, modifier]
2  #heuristic equal(route((T,T'),R,R')). [value, modifier]
3  #heuristic equal(bind(T,R)). [sign, 1]
4  #heuristic equal(route((T,T'),R,R')). [sign, 1]
```

**Listing 19.** Heuristics encouraging `equal/1` atoms.

```
1  #heuristic unequal(bind(T,R)). [value, modifier]
2  #heuristic unequal(route((T,T'),R,R')). [value, modifier]
3  #heuristic unequal(bind(T,R)). [sign, -1]
4  #heuristic unequal(route((T,T'),R,R')). [sign, -1]
```

## 6. Experiments

In this section, we empirically analyze our evolutionary design space exploration techniques. Two main questions are guiding the experiments:

1. Are the solutions found in a short amount of time similar to the parent implementation when using similarity techniques?
2. Is the quality of the solutions found in a short amount of time better when using similarity information?

The first question relies on the Hamming distance discussed in Section 3; note that we only consider the binding and routing distance. The second question is measured by the so-called $\epsilon$-*dominance* [15]. Essentially, a reference best-known Pareto front is built among the results of all experiments, and then an individual approximate Pareto front for a single technique is measured on a scale of zero to one. The higher the value, the closer the approximate Pareto front is to the reference Pareto front.

### 6.1. Experimental Setup

We tested our techniques on 35 instances generated by an ASP-based system [16]. The hardware platform of each instance has a network-on-chip architecture (NoC), as in our running example. The grid size ranges from $3 \times 3 \times 1$, so from 9 routers and 9 computational or memory resources to $3 \times 3 \times 3$, resulting in 27 routers and, accordingly, many computational and memory resources. The applications range from 6 to 160 tasks and are designed to follow a serial-parallel graph pattern. The dependency graphs are not random but are composed of a certain amount of serial patterns, i.e., tasks that are sequentially executed, and parallel patterns, i.e., tasks that can be executed in parallel. Note that the complexity and size of most of these instances is large enough that an exhaustive exact design space exploration is impossible.

Our experimental setup requires two distinct steps:

1. Perform an extensive Pareto optimization on the instance set and select a non-dominated solution as the parent implementation;
2. Obtain child specifications by slightly changing the instances and perform a low-timeout Pareto optimization with and without our various similarity techniques.

For all experiments, a reimplementation of the ASP-based system in [2] in Python was used. Essentially, ASP is combined with two background theories, one allowing for difference constraints, and one handling the Pareto optimization. While enumerating solutions, we check compliance with the difference logic and continuously update an archive of best known solutions, i.e., dominated solutions are removed and better or incomparable ones are added until no more solutions can be found, making the final archive the exact Pareto front. Note that we can exclude partial non-dominated solutions due to the assignment-monotone nature of our objective values, i.e., when more is assigned, the value can only get worse. The system is based on *clingo* 5.5, its extension *clingo*[DL] 1.3, and Python 3.9. To supervise and control the execution, the tool *runlim* version 2.0.0rc3 was

used (*runlim* is available at https://github.com/arminbiere/runlim (accessed on 29 April 2022)).

The first step of the experiments was executed on a machine with a Core Xeon E3-1260Lv5 CPU and 32 GiB of RAM with Linux Debian 10; the runtime and memory were limited to 12 h and 20 GiB, respectively. All three routing techniques are used on the 35 instances, and the long runtime aims to obtain the best possible solutions. As mentioned above, only 6 instances could be completely explored. For all 35 instances, one non-dominated solution, i.e., a solution among the best that any routing technique could find, was selected at random as the parent implementation, and these solutions were reified as described in Section 5.1. Then, 35 child specifications were obtained by randomly changing 20% of applications and 20% of the hardware platform; the changes included the removal, addition, or exchange of the respective specification. Note that we retain the NoC-architecture.

The parent implementation, child specification, and our various encoding techniques were then combined and executed on a machine with an Intel Core Xeon E5-2650v4 CPU and 64 GiB of RAM with Linux Debian 10, where each individual run was limited to 900 s runtime and 20 GiB memory.

In the following evaluation, we refer to the various techniques as follows:

- Routing variants:

  ARB arbitrary routing (Listing 3)
  BOU bound routing (Listings 3 and 8)
  XYZ dimension-ordered routing (Listing 9)

- Similarity techniques:

  – Strategies

    S1 forbid unequal implementations (Listing 14)
    S2 enforce equal implementations (Listing 15)

  – Preferences

    P1 discourage unequal implementations (Listing 16)
    P2 encourage equal implementations (Listing 17)

  – Heuristics

    H1 heuristics discouraging unequal implementations (Listing 18)
    H2 heuristics encouraging equal implementations (Listing 19)
    ∗ additional modifiers for importance are F (factor) with value 2,4, and 8, and L (level) with value 1

- Part of implementation:

  B binding only
  BR binding plus routing

In total, we tested 87 configurations, including the three baseline configurations without any similarity information. The system, all encodings, and the instances set, including the parent and child specifications and the parent implementation, can be found here https://github.com/krr-up/asp-dse/releases/tag/v1.0.1 (accessed on 7 February 2023).

*6.2. Experimental Evaluation*

We begin by highlighting which routing and similarity techniques produce solutions to the instances.

Table 1 shows how many instances were completely solved (SAT), i.e., the Pareto front was found, how many solutions were found but the timeout was reached (SAT+TO), i.e., an approximate Pareto front was found, and how many were unsatisfiable (UNSAT). The remaining instances either could not be grounded or solving yielded no solution. We select the specific configuration for each class of similarity techniques, viz. strategies (S),

preferences (P), heuristics (H), that were able to produce the most solutions and compare them with the baseline configurations. The baseline configurations are highlighted in red.

**Table 1.** Best results regarding instances solved for routing and similarity techniques.

|       | SAT | SAT+TO | UNSAT |
|-------|-----|--------|-------|
| ARB   | 1   | 14     | 0     |
| ARB-S | 1   | 11     | 9     |
| ARB-P | 1   | 6      | 0     |
| ARB-H | 1   | 22     | 0     |
| BOU   | 4   | 23     | 0     |
| BOU-S | 5   | 12     | 18    |
| BOU-P | 4   | 13     | 0     |
| BOU-H | 4   | 27     | 0     |
| XYZ   | 6   | 27     | 0     |
| XYZ-S | 9   | 8      | 18    |
| XYZ-P | 6   | 19     | 0     |
| XYZ-H | 6   | 28     | 0     |

Overall, we clearly see that the approximate routing techniques drastically increase the performance compared with ARB. The XYZ routing performs the best, but it is only slightly better than BOU. This is encouraging, as XYZ is the least flexible routing and therefore might discard high-quality solutions. Among the similarity techniques, we observe the expected phenomena that strategies lead to unsatisfiability and are merely able to produce results for the least instances. For the full routing, the proof of unsatisfiability finished for 9 instances, while with the advanced routing techniques, 18 instances proved to be unsatisfiable, which is a bit more than half of the instance set. For the advanced routing techniques, strategies actually fully explored more instances than any other configuration. Because strategies exclude solutions, the Pareto front of the fully explored instances might be of worse quality. On the other hand, strategies could be a fast technique with guaranteed similarity to satisfiable instances; we return to this below. Preferences performed worse than the baseline. This might be due to the fact that a more complex Pareto optimization is used that includes similarity. For all routing techniques, heuristics supply the best results in terms of instances with solutions, and they are on par with the baseline regarding fully explored instances.

In the following, we use the average rank to compare the different techniques. In essence, each instance induces an ordering among the 87 configurations regarding the measurements of $\epsilon$-dominance and (maximum and average) Hamming distance, along with the product of the two. In the case of a tie, the next entry that is worse receives the rank that it would have been at if no tie occurred; for instance, if two configurations are tied for first, the next configuration gets rank three. These ranks are averaged to provide insights into which configurations performed well overall. Configurations that did not yield any solution to an instance are punished with the worst possible rank. Furthermore, due to the vast amount of configurations, we only present the top 50 results plus the baseline configurations (Less than 53 results are shown whenever the baseline configurations are among the top 50).

Next, we analyze the first solution that was found with respect to similarity and quality. Our hypothesis is that similarity techniques immediately yield not only similar but better solutions compared with the baseline, as the parent implementation is of high quality.

Table 2 shows from left to right the top 50 configurations for Hamming distance (HD), $\epsilon$-dominance ($\epsilon$D), and their product (HD×$\epsilon$D).

**Table 2.** Average rank regarding Hamming distance, $\epsilon$-dominance, and their product for the first solution found.

| | HD | | $\epsilon$D | | HD$\times\epsilon$D |
|---|---|---|---|---|---|
| **C** | **AVG-R** | **C** | **AVG-R** | **C** | **AVG-R** |
| BOU-BR-H1-L | 8.3 | XYZ-BR-H2 | 17.4 | BOU-BR-H2-L | 14.1 |
| BOU-BR-H2-L | 8.5 | XYZ-B-H1-L | 18.1 | BOU-BR-H1-F-8 | 14.3 |
| BOU-BR-H1-F-8 | 10.5 | XYZ-B-H2-L | 18.1 | BOU-BR-H1-L | 14.3 |
| BOU-BR-H2-F-8 | 10.8 | XYZ-BR-H2-F-4 | 19.2 | BOU-BR-H1-F-2 | 14.4 |
| BOU-BR-H1-F-2 | 12.3 | XYZ-BR-H2-F-2 | 19.4 | BOU-BR-H2-F-8 | 14.8 |
| BOU-BR-H2-F-4 | 13.0 | XYZ-BR-H2-L | 19.4 | BOU-BR-H2-F-2 | 16.7 |
| BOU-BR-H1-F-4 | 13.4 | XYZ-BR-H1 | 19.8 | XYZ-B-H1-L | 17.1 |
| BOU-BR-H2-F-2 | 13.7 | XYZ-BR-H1-F-2 | 19.9 | XYZ-B-H2-L | 17.1 |
| XYZ-BR-H1-L | 14.9 | XYZ-BR-H2-F-8 | 20.0 | BOU-BR-H1-F-4 | 17.9 |
| XYZ-B-H1-L | 15.3 | BOU-BR-H1-F-8 | 20.2 | BOU-BR-H2-F-4 | 18.0 |
| XYZ-B-H2-L | 15.3 | XYZ-BR-H1-F-8 | 20.3 | XYZ-BR-H1-L | 18.8 |
| XYZ-BR-H2-L | 17.1 | BOU-B-H1-F-4 | 20.4 | XYZ-BR-H2-L | 18.9 |
| BOU-B-H1-L | 18.7 | BOU-B-H2-F-4 | 20.4 | XYZ-BR-H1-F-8 | 21.3 |
| BOU-B-H2-L | 18.7 | XYZ-BR-H1-F-4 | 20.4 | XYZ-BR-H2-F-8 | 21.3 |
| XYZ-BR-H1-F-8 | 19.6 | XYZ-BR-H1-L | 20.7 | BOU-BR-H1 | 21.6 |
| XYZ-BR-H2-F-8 | 20.8 | BOU-BR-H2-F-8 | 21.0 | XYZ-BR-H2-F-2 | 21.9 |
| XYZ-BR-H1-F-4 | 21.0 | XYZ-B-H1-F-8 | 21.2 | XYZ-BR-H1-F-4 | 22.4 |
| BOU-BR-H1 | 22.1 | XYZ-B-H2-F-8 | 21.2 | BOU-BR-H2 | 22.7 |
| BOU-BR-H2 | 22.1 | BOU-BR-H1-F-2 | 21.4 | BOU-B-H1-F-4 | 22.7 |
| ARB-BR-H2-L | 23.1 | BOU-BR-H1-L | 21.7 | BOU-B-H2-F-4 | 22.7 |
| BOU-B-H1-F-8 | 23.4 | BOU-B-H1-F-2 | 21.9 | XYZ-BR-H2-F-4 | 22.9 |
| BOU-B-H2-F-8 | 23.4 | BOU-B-H2-F-2 | 21.9 | XYZ-BR-H1-F-2 | 23.0 |
| XYZ-B-H1-F-4 | 23.4 | BOU-BR-H2-L | 22.0 | BOU-B-H1-L | 23.3 |
| XYZ-B-H2-F-4 | 23.4 | <span style="color:red">XYZ</span> | 22.3 | BOU-B-H2-L | 23.3 |
| XYZ-BR-H2-F-4 | 23.4 | BOU-BR-H1 | 22.3 | XYZ-BR-H2 | 23.5 |
| XYZ-B-H1-F-8 | 23.5 | BOU-B-H1-F-8 | 22.5 | XYZ-B-H1-F-8 | 23.8 |
| XYZ-B-H2-F-8 | 23.5 | BOU-B-H2-F-8 | 22.5 | XYZ-B-H2-F-8 | 23.8 |
| XYZ-BR-H2-F-2 | 23.8 | BOU-B-H1-L | 23.1 | XYZ-BR-H1 | 24.1 |
| ARB-BR-H1-L | 24.0 | BOU-B-H2-L | 23.1 | BOU-B-H1-F-8 | 24.3 |
| XYZ-BR-H1-F-2 | 25.5 | BOU-BR-H2 | 23.4 | BOU-B-H2-F-8 | 24.3 |
| BOU-B-H1-F-4 | 25.6 | XYZ-B-H1-F-2 | 23.6 | XYZ-B-H1-F-4 | 25.1 |
| BOU-B-H2-F-4 | 25.6 | XYZ-B-H2-F-2 | 23.6 | XYZ-B-H2-F-4 | 25.1 |
| XYZ-B-H1-F-2 | 25.7 | XYZ-B-H1-F-4 | 23.9 | BOU-B-H1-F-2 | 25.3 |
| XYZ-B-H2-F-2 | 25.7 | XYZ-B-H2-F-4 | 23.9 | BOU-B-H2-F-2 | 25.3 |
| BOU-B-H1-F-2 | 26.3 | BOU-BR-H2-F-2 | 23.9 | XYZ-B-H1-F-2 | 25.9 |
| BOU-B-H2-F-2 | 26.3 | BOU-B-H1 | 24.2 | XYZ-B-H2-F-2 | 25.9 |
| XYZ-BR-H1 | 28.6 | BOU-B-H2 | 24.2 | BOU-B-H1 | 28.9 |
| XYZ-BR-H2 | 29.7 | BOU-BR-H2-F-4 | 24.3 | BOU-B-H2 | 28.9 |
| XYZ-B-S2 | 29.8 | BOU-BR-H1-F-4 | 24.5 | XYZ-B-H1 | 29.6 |
| BOU-B-S2 | 31.5 | XYZ-B-H1 | 25.3 | XYZ-B-H2 | 29.6 |
| ARB-BR-H2-F-8 | 31.6 | XYZ-B-H2 | 25.3 | XYZ-B-S2 | 30.0 |
| ARB-BR-H1-F-8 | 31.8 | <span style="color:red">BOU</span> | 25.3 | BOU-B-S2 | 32.2 |
| XYZ-B-H1 | 31.8 | XYZ-B-P2 | 26.2 | ARB-BR-H1-L | 34.3 |
| XYZ-B-H2 | 31.8 | XYZ-B-S2 | 30.1 | <span style="color:red">XYZ</span> | 34.6 |
| ARB-BR-H2-F-4 | 32.6 | XYZ-B-P1 | 31.6 | ARB-BR-H2-L | 35.2 |
| BOU-B-H1 | 34.0 | BOU-B-S2 | 31.6 | <span style="color:red">BOU</span> | 35.6 |
| BOU-B-H2 | 34.0 | XYZ-BR-P1 | 33.1 | ARB-BR-H2-F-4 | 36.5 |
| ARB-BR-H1-F-4 | 34.5 | XYZ-BR-P2 | 33.1 | ARB-BR-H2-F-8 | 37.1 |
| ARB-BR-H2-F-2 | 35.9 | BOU-B-P2 | 33.8 | ARB-BR-H1-F-8 | 37.2 |
| ARB-BR-H1-F-2 | 37.6 | BOU-B-P1 | 34.5 | XYZ-B-P2 | 37.3 |
| <span style="color:red">BOU</span> | 43.2 | <span style="color:red">ARB</span> | 42.5 | <span style="color:red">ARB</span> | 51.1 |
| <span style="color:red">XYZ</span> | 43.3 | | | | |
| <span style="color:red">ARB</span> | 52.2 | | | | |

Regarding Hamming distance, most configurations using heuristics or S2 with advanced routing and binding distance improve over the baselines. Routing BOU with

enhanced importance of similarity atoms clearly performs best; the more emphasis on deciding similarity atoms first, the closer the first answer. This may be due to more flexibility of BOU routing compared with XYZ, and the reordering of the search space by the heuristic does not seem to negatively impact the solving. There is no clear favorite between heuristics H1 and H2, but taking both binding and routing similarity into account is at the top. Because S1 and preferences overall do not find as many solutions, they are absent from the top 50.

With respect to $\epsilon$-dominance, i.e., the quality of the first solution, advanced routing techniques and heuristics are again at the top. Now, the XYZ routing is in front, likely due to the smaller problem size and search space allowing for a faster exploration, even if it is approximate. Note that the spread of ranks is a lot closer among all configurations; this indicates that the top configurations found first solutions of similar quality.

Finally, BOU routing with heuristics and both binding and routing similarities achieves the best ranks by combining similarity and quality, and it also outperforms the baselines. The trend that could be seen for individual Hamming distance and $\epsilon$-dominance continues. It seems that the restrictive strategies and more complex preferences do not provide a good trade-off. On the other hand, heuristics and advanced routing clearly outperforms the design space exploration from scratch.

Table 3 provides the top 50 ranking for the average (AVG-HD) and maximum Hamming distance (MAX-HD) among the final solution set. The picture is almost identical to the first solution. Routing BOU with heuristics that puts heavy emphasis on deciding similarities first clearly outperforms the other configurations. Similarly, the only non-heuristic technique that is successful is S2 with binding distance only. The average and maximum also behave very similarly.

Table 4 shows the $\epsilon$-dominance ($\epsilon$D) and the product of $\epsilon$-dominance and Hamming distance (HD$\times\epsilon$D) for the final solution set. Now, XYZ routing again outperforms the other techniques. Similarly, heuristics with more importance on deciding similarities first are at the very top. Moreover, the approximate but efficient routing can explore more solutions over time and therefore ultimately collects solutions of higher quality. The high performance of the XYZ routing and the similarity achieved by the BOU routing plus similarity through heuristics leads to these techniques being at the top of the combined measure for quality and similarity, with routing BOU plus H with L ranking slightly ahead. Overall, those configurations achieve our initial goal of high-quality solutions that are similar to the parent implementation within a short amount of time.

In addition to the more complex optimization, the failure of preferences can be explained by the fact that extreme solutions might be very similar but of low quality or have good quality and low similarity. If the latter is found first, it is non-dominated but not desired. This might explain why preferences do not occur in the top 50 for similarity but occur for $\epsilon$-dominance. On the other hand, strategies occur in both and might still be a good candidate for solutions that can be satisfied; we explore this in more detail next. To analyze the viability of strategies, we restrict the analysis to the 17 instances for which strategies found solutions.

Table 5 provides the same information as Table 2 for the 17 instances and highlights successful strategies in bold. Now, we see that S2 for binding only with advances routing techniques ranks among the top 20 for Hamming distance, first and third in regards to $\epsilon$-dominance, and 8th and 12th for the combined measure, respectively. Note that this strategy guarantees that the same bindings are chosen from the parent implementation, but it does allow newly added mappings to be used as well. Overall, the same heuristic techniques still perform better, but for some instances, strategies may be applicable while offering guarantees.

Table 6 relays the same information as Table 3 for the restricted instance set. Again, the highlighted strategy is only outperformed by the heuristic that also takes into account the routing distance. For the average Hamming distance, S2 achieves rank 9 and 11, and for the maximum Hamming distance, it ranks 13 and 14.

Table 7 is the analog to Table 4. For the final solution set, strategies are still in the top 50 with respect to quality, but baseline and heuristic techniques achieve higher quality for their respective advanced routing technique. As before, the more effective XYZ routing is first followed by BOU routing. Recall that strategies restrict the search space and therefore might discard high-quality answers for the child specification. The combined measure still has both advanced routing variants in the top 50 and above all baselines, with XYZ-B-S2 at rank 10 and BOU-B-S2 at 37.

**Table 3.** Average rank regarding average and maximum Hamming distance after cutoff time.

| | **AVG-HD** | | **MAX-HD** |
|---|---|---|---|
| **C** | **AVG-R** | **C** | **AVG-R** |
| BOU-BR-H1-L | 9.7 | BOU-BR-H1-L | 8.4 |
| BOU-BR-H2-L | 9.9 | BOU-BR-H2-L | 8.6 |
| BOU-BR-H1-F-8 | 10.9 | BOU-BR-H1-F-8 | 10.4 |
| BOU-BR-H2-F-8 | 11.3 | BOU-BR-H2-F-8 | 10.7 |
| BOU-BR-H2-F-4 | 12.4 | BOU-BR-H2-F-4 | 12.1 |
| BOU-BR-H1-F-4 | 13.0 | BOU-BR-H1-F-4 | 12.7 |
| BOU-BR-H1-F-2 | 14.6 | BOU-BR-H1-F-2 | 13.3 |
| BOU-BR-H2-F-2 | 15.9 | BOU-BR-H2-F-2 | 14.7 |
| BOU-B-H1-L | 18.6 | XYZ-BR-H1-L | 16.0 |
| BOU-B-H2-L | 18.6 | XYZ-B-H2-L | 16.8 |
| ARB-BR-H2-L | 20.1 | XYZ-B-H1-L | 16.9 |
| XYZ-BR-H1-L | 20.5 | BOU-B-H1-L | 17.5 |
| ARB-BR-H1-L | 20.7 | BOU-B-H2-L | 17.5 |
| XYZ-B-H2-L | 21.0 | XYZ-BR-H2-L | 17.8 |
| XYZ-B-H1-L | 21.1 | XYZ-BR-H1-F-8 | 20.3 |
| BOU-BR-H1 | 22.2 | BOU-BR-H2 | 20.5 |
| BOU-BR-H2 | 22.3 | BOU-BR-H1 | 20.6 |
| XYZ-BR-H2-L | 22.8 | BOU-B-H1-F-8 | 20.7 |
| BOU-B-H1-F-8 | 22.8 | BOU-B-H2-F-8 | 20.7 |
| BOU-B-H2-F-8 | 22.8 | ARB-BR-H2-L | 21.7 |
| XYZ-BR-H1-F-8 | 25.1 | BOU-B-H1-F-4 | 22.1 |
| BOU-B-H1-F-4 | 25.9 | BOU-B-H2-F-4 | 22.1 |
| BOU-B-H2-F-4 | 25.9 | ARB-BR-H1-L | 22.4 |
| BOU-B-H1-F-2 | 26.0 | XYZ-BR-H2-F-8 | 22.9 |
| BOU-B-H2-F-2 | 26.0 | XYZ-B-H1-F-4 | 23.2 |
| XYZ-B-S2 | 27.0 | XYZ-B-H2-F-4 | 23.2 |
| XYZ-B-H1-F-4 | 27.4 | XYZ-BR-H1-F-2 | 23.4 |
| XYZ-B-H2-F-4 | 27.4 | BOU-B-H1-F-2 | 23.5 |
| BOU-B-S2 | 27.6 | BOU-B-H2-F-2 | 23.5 |
| XYZ-BR-H2-F-8 | 27.8 | XYZ-BR-H2-F-4 | 23.6 |
| XYZ-BR-H1-F-4 | 28.2 | XYZ-BR-H1-F-4 | 23.7 |
| XYZ-BR-H1-F-2 | 28.3 | XYZ-BR-H2-F-2 | 24.2 |
| XYZ-BR-H2-F-4 | 28.6 | XYZ-B-H1-F-8 | 24.9 |
| ARB-BR-H1-F-8 | 28.9 | XYZ-B-H2-F-8 | 24.9 |
| XYZ-B-H1-F-8 | 28.9 | XYZ-B-H1-F-2 | 26.1 |
| ARB-BR-H2-F-8 | 28.9 | XYZ-B-H2-F-2 | 26.1 |
| XYZ-B-H2-F-8 | 28.9 | BOU-B-H1 | 26.5 |
| XYZ-BR-H2-F-2 | 29.5 | BOU-B-H2 | 26.5 |
| BOU-B-H1 | 29.7 | XYZ-BR-H1 | 28.2 |
| BOU-B-H2 | 29.7 | XYZ-BR-H2 | 28.7 |
| XYZ-B-H2-F-2 | 30.4 | XYZ-B-H2 | 29.0 |
| XYZ-B-H1-F-2 | 30.4 | XYZ-B-H1 | 29.1 |
| ARB-BR-H2-F-4 | 32.4 | XYZ-B-S2 | 29.5 |
| XYZ-BR-H2 | 33.5 | BOU-B-S2 | 30.1 |
| XYZ-BR-H1 | 33.6 | ARB-BR-H1-F-8 | 30.2 |
| ARB-BR-H1-F-4 | 34.1 | ARB-BR-H2-F-8 | 30.3 |
| XYZ-B-H2 | 34.2 | ARB-BR-H2-F-4 | 32.4 |
| XYZ-B-H1 | 34.3 | ARB-BR-H1-F-4 | 34.2 |
| ARB-BR-H2-F-2 | 35.1 | XYZ | 35.8 |
| ARB-BR-H1-F-2 | 36.5 | ARB-BR-H2-F-2 | 36.1 |
| XYZ | 41.1 | BOU | 38.5 |
| BOU | 42.5 | ARB | 50.5 |
| ARB | 50.5 | | |

**Table 4.** Average rank regarding $\epsilon$-dominance and the product with average Hamming distance after cutoff time.

| | $\epsilon$D | | $HD \times \epsilon D$ |
|---|---|---|---|
| **C** | **AVG-R** | **C** | **AVG-R** |
| XYZ-B-H1-L | 3.9 | BOU-BR-H1-L | 12.1 |
| XYZ-B-H2-L | 3.9 | BOU-BR-H2-L | 13.1 |
| XYZ-BR-H1-L | 4.7 | XYZ-B-H2-L | 13.6 |
| XYZ-BR-H2-F-8 | 6.1 | XYZ-B-H1-L | 13.7 |
| XYZ-BR-H1-F-2 | 6.5 | XYZ-BR-H1-L | 14.4 |
| XYZ-BR-H2-L | 6.6 | BOU-BR-H1-F-8 | 15.9 |
| XYZ-BR-H2-F-4 | 6.9 | XYZ-BR-H2-L | 16.9 |
| XYZ-BR-H1-F-8 | 6.9 | XYZ-BR-H1-F-8 | 17.8 |
| XYZ-BR-H2-F-2 | 7.1 | BOU-BR-H2-F-8 | 18.1 |
| XYZ-BR-H1-F-4 | 7.3 | XYZ-BR-H1-F-2 | 18.8 |
| XYZ-B-H2-F-4 | 8.0 | XYZ-B-H2-F-4 | 19.1 |
| XYZ-B-H1-F-4 | 8.1 | XYZ-BR-H1-F-4 | 19.2 |
| XYZ-BR-H2 | 8.5 | XYZ-B-H1-F-4 | 19.4 |
| XYZ-B-H1-F-2 | 9.3 | XYZ-BR-H2-F-4 | 19.5 |
| XYZ-B-H2-F-2 | 9.3 | XYZ-BR-H2-F-8 | 19.8 |
| XYZ | 9.5 | BOU-BR-H1-F-4 | 20.3 |
| XYZ-B-H1-F-8 | 10.1 | BOU-BR-H1-F-2 | 20.6 |
| XYZ-B-H2-F-8 | 10.1 | XYZ-BR-H2-F-2 | 20.6 |
| XYZ-B-H1 | 10.7 | XYZ-B-H1-F-8 | 20.7 |
| XYZ-B-H2 | 10.7 | XYZ-B-H2-F-8 | 20.7 |
| XYZ-BR-H1 | 11.3 | BOU-BR-H2-F-4 | 20.8 |
| XYZ-B-P2 | 13.0 | XYZ-B-H2-F-2 | 21.8 |
| BOU-BR-H1-L | 14.8 | XYZ-B-H1-F-2 | 21.8 |
| BOU-BR-H2-L | 15.5 | XYZ-BR-H2 | 22.7 |
| BOU-B-H1-F-4 | 19.1 | BOU-BR-H2-F-2 | 23.1 |
| BOU-B-H2-F-4 | 19.1 | XYZ-BR-H1 | 24.3 |
| BOU-BR-H1-F-8 | 19.6 | BOU-BR-H2 | 24.4 |
| BOU-BR-H2-F-8 | 20.7 | BOU-BR-H1 | 24.7 |
| XYZ-B-P1 | 21.1 | BOU-B-H1-F-4 | 26.0 |
| BOU-B-H1-F-2 | 21.6 | BOU-B-H2-F-4 | 26.0 |
| BOU-B-H2-F-2 | 21.6 | XYZ-B-H2 | 26.1 |
| XYZ-BR-P1 | 22.3 | BOU-B-H1-L | 26.2 |
| XYZ-BR-P2 | 22.3 | BOU-B-H2-L | 26.2 |
| BOU-BR-H1 | 22.3 | XYZ-B-H1 | 26.2 |
| BOU | 22.4 | BOU-B-H1-F-2 | 26.9 |
| BOU-BR-H2 | 22.6 | BOU-B-H2-F-2 | 26.9 |
| BOU-B-H1-L | 22.8 | BOU-B-H1-F-8 | 27.0 |
| BOU-B-H2-L | 22.8 | BOU-B-H2-F-8 | 27.0 |
| BOU-BR-H1-F-4 | 22.8 | BOU-B-H1 | 29.0 |
| BOU-BR-H1-F-2 | 23.1 | BOU-B-H2 | 29.1 |
| BOU-B-H1-F-8 | 23.4 | XYZ-B-S2 | 30.9 |
| BOU-B-H2-F-8 | 23.4 | XYZ | 32.0 |
| BOU-B-H1 | 23.5 | ARB-BR-H2-L | 33.4 |
| BOU-B-H2 | 23.5 | ARB-BR-H1-L | 33.5 |
| BOU-BR-H2-F-4 | 24.2 | XYZ-B-P2 | 35.5 |
| BOU-BR-H2-F-2 | 25.7 | BOU-B-S2 | 35.7 |
| BOU-B-P2 | 29.2 | BOU | 37.1 |
| BOU-B-P1 | 29.3 | XYZ-B-P1 | 37.5 |
| BOU-BR-P1 | 29.3 | XYZ-BR-P1 | 38.1 |
| BOU-BR-P2 | 30.4 | ARB-BR-H1-F-8 | 39.2 |
| ARB | 44.3 | ARB | 50.1 |

**Table 5.** Average rank regarding Hamming distance, $\epsilon$-dominance, and their product for the first solution found with instances where strategies were applicable.

| C | HD AVG-R | C | $\epsilon$D AVG-R | C | HD$\times\epsilon$D AVG-R |
|---|---|---|---|---|---|
| BOU-BR-H1-F-8 | 6.9 | **XYZ-B-S2** | 17.2 | BOU-BR-H1-F-2 | 12.2 |
| BOU-BR-H2-L | 7.4 | XYZ-BR-H2 | 17.8 | BOU-BR-H2-L | 12.8 |
| BOU-BR-H2-F-8 | 7.7 | **BOU-B-S2** | 20.2 | BOU-BR-H1-F-8 | 13.2 |
| BOU-BR-H1-L | 8.1 | BOU-BR-H1 | 20.2 | BOU-BR-H1-L | 13.6 |
| BOU-BR-H1-F-2 | 8.4 | BOU-BR-H1-F-2 | 21.1 | BOU-BR-H2-F-8 | 14.0 |
| BOU-BR-H2-F-4 | 9.6 | BOU-B-H1-F-4 | 21.2 | BOU-BR-H2-F-4 | 14.5 |
| ARB-BR-H2-L | 10.4 | BOU-B-H2-F-4 | 21.2 | BOU-BR-H1-F-4 | 15.4 |
| BOU-BR-H1-F-4 | 10.5 | XYZ-BR-H1 | 21.8 | **XYZ-B-S2** | 16.9 |
| BOU-BR-H2-F-2 | 11.4 | BOU-B-H1-F-2 | 21.9 | BOU-BR-H2-F-2 | 17.4 |
| ARB-BR-H1-L | 12.1 | BOU-B-H2-F-2 | 21.9 | BOU-BR-H1 | 18.0 |
| **XYZ-B-S2** | 16.5 | XYZ-BR-H1-F-8 | 22.1 | BOU-BR-H2 | 20.5 |
| BOU-BR-H1 | 18.7 | BOU-B-H1-F-8 | 22.5 | **BOU-B-S2** | 21.4 |
| BOU-BR-H2 | 18.8 | BOU-B-H2-F-8 | 22.5 | XYZ-BR-H2-L | 23.4 |
| XYZ-B-H1-L | 19.8 | BOU-BR-H1-F-8 | 22.8 | XYZ-B-H1-L | 23.5 |
| XYZ-B-H2-L | 19.8 | BOU-BR-H1-L | 22.9 | XYZ-B-H2-L | 23.5 |
| **BOU-B-S2** | 20.0 | BOU-BR-H2 | 23.1 | XYZ-BR-H2 | 24.6 |
| BOU-B-H1-L | 20.9 | XYZ-BR-H1-F-2 | 23.1 | XYZ-BR-H1-L | 24.8 |
| BOU-B-H2-L | 20.9 | XYZ-BR-H1-F-4 | 23.3 | BOU-B-H1-F-8 | 25.4 |
| XYZ-BR-H1-L | 21.3 | XYZ-B-H1-L | 23.6 | BOU-B-H2-F-8 | 25.4 |
| XYZ-BR-H2-L | 21.8 | XYZ-B-H2-L | 23.6 | BOU-B-H1-F-4 | 25.6 |
| XYZ-BR-H1-F-8 | 26.1 | BOU-BR-H2-L | 23.8 | BOU-B-H2-F-4 | 25.6 |
| BOU-B-H1-F-8 | 26.3 | BOU-BR-H2-F-4 | 23.8 | XYZ-BR-H1-F-8 | 25.8 |
| BOU-B-H2-F-8 | 26.3 | XYZ-BR-H2-F-4 | 23.9 | XYZ-BR-H1-F-4 | 27.0 |
| ARB-BR-H2-F-8 | 28.0 | BOU-BR-H2-F-8 | 24.1 | BOU-B-H1-F-2 | 28.0 |
| XYZ-BR-H1-F-4 | 28.1 | XYZ-BR-H2-F-2 | 24.4 | BOU-B-H2-F-2 | 28.0 |
| XYZ-BR-H2-F-8 | 28.2 | XYZ-BR-H2-L | 24.5 | XYZ-BR-H1 | 28.1 |
| ARB-BR-H1-F-8 | 28.4 | XYZ-BR-H2-F-8 | 25.0 | XYZ-BR-H1-F-2 | 28.4 |
| ARB-BR-H2-F-4 | 29.6 | XYZ-B-H1 | 25.2 | XYZ-BR-H2-F-8 | 28.5 |
| BOU-B-H1-F-4 | 30.1 | XYZ-B-H2 | 25.2 | BOU-B-H1-L | 28.5 |
| BOU-B-H2-F-4 | 30.1 | BOU-BR-H1-F-4 | 25.4 | BOU-B-H2-L | 28.5 |
| XYZ-BR-H2-F-4 | 30.8 | XYZ-BR-H1-L | 25.4 | XYZ-BR-H2-F-4 | 29.4 |
| XYZ-BR-H1-F-2 | 32.0 | <span style="color:red">BOU</span> | 26.5 | XYZ-BR-H2-F-2 | 29.5 |
| XYZ-B-H1-F-8 | 32.1 | BOU-BR-H2-F-2 | 27.5 | ARB-BR-H1-L | 30.6 |
| XYZ-B-H2-F-8 | 32.1 | BOU-B-H1 | 27.5 | XYZ-B-H1-F-2 | 31.8 |
| XYZ-B-H1-F-4 | 32.5 | BOU-B-H2 | 27.5 | XYZ-B-H2-F-2 | 31.8 |
| XYZ-B-H2-F-4 | 32.5 | XYZ-B-H1-F-2 | 27.8 | ARB-BR-H2-L | 32.5 |
| BOU-B-H1-F-2 | 32.6 | XYZ-B-H2-F-2 | 27.8 | XYZ-B-H1-F-8 | 33.1 |
| BOU-B-H2-F-2 | 32.6 | XYZ-B-H1-F-8 | 28.0 | XYZ-B-H2-F-8 | 33.1 |
| XYZ-BR-H2-F-2 | 32.6 | XYZ-B-H2-F-8 | 28.0 | XYZ-B-H1-F-4 | 33.4 |
| XYZ-B-H1-F-2 | 32.6 | <span style="color:red">XYZ</span> | 28.1 | XYZ-B-H2-F-4 | 33.4 |
| XYZ-B-H2-F-2 | 32.6 | BOU-B-H1-L | 29.1 | BOU-B-H1 | 33.5 |
| ARB-BR-H1-F-4 | 33.5 | BOU-B-H2-L | 29.1 | BOU-B-H2 | 33.5 |
| ARB-BR-H2-F-2 | 34.3 | XYZ-B-H1-F-4 | 30.0 | XYZ-B-H1 | 33.7 |
| XYZ-BR-H1 | 34.4 | XYZ-B-H2-F-4 | 30.0 | XYZ-B-H2 | 33.7 |
| XYZ-BR-H2 | 34.7 | XYZ-B-P2 | 31.5 | ARB-BR-H2-F-8 | 36.4 |
| ARB-BR-H1-F-2 | 37.8 | ARB-BR-H1-L | 34.8 | ARB-BR-H1-F-8 | 36.5 |
| XYZ-B-H1 | 39.2 | ARB-BR-H1-F-8 | 35.8 | ARB-BR-H2-F-4 | 37.1 |
| XYZ-B-H2 | 39.2 | ARB-BR-H2-F-8 | 36.2 | ARB-BR-H1-F-4 | 39.1 |
| BOU-B-H1 | 40.2 | ARB-BR-H2-L | 36.7 | <span style="color:red">BOU</span> | 40.5 |
| BOU-B-H2 | 40.2 | ARB-BR-H2-F-4 | 38.1 | ARB-BR-H2-F-2 | 41.1 |
| <span style="color:red">BOU</span> | 52.8 | <span style="color:red">ARB</span> | 46.5 | <span style="color:red">XYZ</span> | 45.9 |
| <span style="color:red">XYZ</span> | 56.5 | | | <span style="color:red">ARB</span> | 61.8 |
| <span style="color:red">ARB</span> | 63.8 | | | | |

**Table 6.** Average rank regarding average and maximum Hamming distance after cutoff time with instances where strategies were applicable. Successful strategies are highlighted in bold.

| | AVG-HD | | MAX-HD |
| --- | --- | --- | --- |
| C | AVG-R | C | AVG-R |
| ARB-BR-H2-L | 9.1 | BOU-BR-H1-F-8 | 7.9 |
| BOU-BR-H1-F-8 | 10.1 | BOU-BR-H2-L | 8.2 |
| ARB-BR-H1-L | 10.2 | BOU-BR-H2-F-8 | 8.3 |
| BOU-BR-H2-L | 10.3 | BOU-BR-H2-F-4 | 8.8 |
| BOU-BR-H2-F-4 | 10.6 | BOU-BR-H1-L | 8.8 |
| BOU-BR-H2-F-8 | 10.8 | BOU-BR-H1-F-2 | 9.5 |
| BOU-BR-H1-F-2 | 11.0 | BOU-BR-H1-F-4 | 9.9 |
| BOU-BR-H1-L | 11.1 | ARB-BR-H2-L | 10.9 |
| **XYZ-B-S2** | 11.4 | ARB-BR-H1-L | 12.2 |
| BOU-BR-H1-F-4 | 11.9 | BOU-BR-H2-F-2 | 12.6 |
| **BOU-B-S2** | 12.6 | BOU-B-H1-L | 15.4 |
| BOU-BR-H2-F-2 | 13.8 | BOU-B-H2-L | 15.4 |
| BOU-B-H1-L | 18.1 | **XYZ-B-S2** | 16.5 |
| BOU-B-H2-L | 18.1 | **BOU-B-S2** | 17.8 |
| BOU-BR-H1 | 22.8 | BOU-B-H1-F-8 | 18.4 |
| BOU-BR-H2 | 22.9 | BOU-B-H2-F-8 | 18.4 |
| BOU-B-H1-F-8 | 23.1 | BOU-BR-H2 | 18.8 |
| BOU-B-H2-F-8 | 23.1 | BOU-BR-H1 | 19.0 |
| ARB-BR-H1-F-8 | 23.8 | XYZ-B-H2-L | 21.0 |
| ARB-BR-H2-F-8 | 23.9 | XYZ-B-H1-L | 21.1 |
| BOU-B-H1-F-2 | 27.6 | BOU-B-H1-F-4 | 22.0 |
| BOU-B-H2-F-2 | 27.6 | BOU-B-H2-F-4 | 22.0 |
| XYZ-B-H2-L | 28.2 | XYZ-BR-H1-L | 22.1 |
| XYZ-B-H1-L | 28.4 | XYZ-BR-H2-L | 22.5 |
| ARB-BR-H2-F-4 | 29.3 | BOU-B-H1-F-2 | 23.4 |
| BOU-B-H2-F-4 | 29.4 | BOU-B-H2-F-2 | 23.4 |
| BOU-B-H1-F-4 | 29.4 | ARB-BR-H1-F-8 | 26.4 |
| XYZ-BR-H1-L | 30.4 | ARB-BR-H2-F-8 | 26.6 |
| XYZ-BR-H2-L | 30.8 | XYZ-BR-H1-F-8 | 28.3 |
| ARB-BR-H1-F-4 | 32.7 | BOU-B-H1 | 28.7 |
| BOU-B-H1 | 33.5 | BOU-B-H2 | 28.7 |
| BOU-B-H2 | 33.5 | ARB-BR-H2-F-4 | 29.2 |
| ARB-BR-H2-F-2 | 34.2 | XYZ-BR-H2-F-8 | 30.2 |
| XYZ-BR-H1-F-8 | 36.4 | XYZ-BR-H1-F-4 | 31.9 |
| ARB-BR-H1-F-2 | 37.1 | ARB-BR-H1-F-4 | 32.9 |
| XYZ-BR-H2-F-8 | 38.2 | XYZ-B-H1-F-4 | 32.9 |
| XYZ-BR-H1-F-4 | 39.4 | XYZ-B-H2-F-4 | 32.9 |
| XYZ-B-H1-F-4 | 40.0 | XYZ-BR-H2-F-4 | 33.1 |
| XYZ-B-H1-F-8 | 40.0 | XYZ-B-H1-F-8 | 33.3 |
| XYZ-B-H2-F-4 | 40.1 | XYZ-B-H2-F-8 | 33.3 |
| XYZ-B-H2-F-8 | 40.1 | XYZ-BR-H2-F-2 | 33.3 |
| XYZ-BR-H2-F-4 | 40.6 | ARB-BR-H2-F-2 | 33.9 |
| XYZ-B-S1 | 41.3 | XYZ-BR-H1-F-2 | 34.2 |
| BOU-B-S1 | 41.5 | XYZ-B-H1-F-2 | 34.8 |
| XYZ-BR-H1-F-2 | 41.8 | XYZ-B-H2-F-2 | 34.8 |
| XYZ-B-H1-F-2 | 42.1 | XYZ-BR-H1 | 35.6 |
| XYZ-B-H2-F-2 | 42.1 | XYZ-BR-H2 | 35.9 |
| XYZ-BR-H2-F-2 | 42.6 | ARB-BR-H1-F-2 | 37.1 |
| XYZ-BR-H2 | 43.1 | XYZ-B-H2 | 37.7 |
| XYZ-BR-H1 | 43.3 | XYZ-B-H1 | 37.9 |
| <span style="color:red">BOU</span> | 50.6 | <span style="color:red">BOU</span> | 44.5 |
| <span style="color:red">XYZ</span> | 54.1 | <span style="color:red">XYZ</span> | 45.5 |
| <span style="color:red">ARB</span> | 60.9 | <span style="color:red">ARB</span> | 60.8 |

**Table 7.** Average rank regarding $\epsilon$-dominance and the product with average Hamming distance after cutoff time with instances where strategies were applicable. Successful strategies are highlighted in bold.

| | $\epsilon$D | | HD $\times$ $\epsilon$D |
|---|---|---|---|
| **C** | **AVG-R** | **C** | **AVG-R** |
| XYZ-BR-H1-L | 3.2 | BOU-BR-H1-L | 6.9 |
| XYZ-B-H1-L | 3.6 | BOU-BR-H2-L | 8.2 |
| XYZ-B-H2-L | 3.6 | BOU-BR-H1-F-8 | 14.2 |
| XYZ-BR-H1-F-8 | 3.8 | XYZ-B-H2-L | 17.6 |
| XYZ-BR-H1-F-2 | 4.9 | XYZ-B-H1-L | 17.6 |
| XYZ-BR-H2-L | 4.9 | BOU-BR-H2-F-8 | 18.1 |
| XYZ-BR-H2-F-8 | 5.9 | BOU-BR-H1-F-2 | 18.3 |
| XYZ-BR-H2-F-4 | 6.6 | BOU-BR-H1-F-4 | 18.5 |
| XYZ-BR-H1-F-4 | 7.2 | BOU-BR-H2-F-4 | 19.2 |
| XYZ-BR-H2 | 8.4 | **XYZ-B-S2** | 19.5 |
| XYZ-B-H1 | 8.8 | XYZ-BR-H1-L | 19.6 |
| XYZ-B-H2-F-4 | 8.8 | XYZ-BR-H2-L | 20.2 |
| XYZ-B-H2 | 8.8 | XYZ-BR-H1-F-8 | 21.2 |
| XYZ-B-H1-F-4 | 8.9 | BOU-BR-H2-F-2 | 23.9 |
| BOU-BR-H1-L | 9.0 | XYZ-BR-H2-F-8 | 24.8 |
| <span style="color:red">XYZ</span> | 9.3 | BOU-BR-H1 | 25.7 |
| BOU-BR-H2-L | 9.5 | XYZ-BR-H1-F-2 | 25.9 |
| XYZ-BR-H2-F-2 | 10.4 | BOU-B-H1-F-2 | 26.0 |
| XYZ-B-H1-F-8 | 12.1 | BOU-B-H2-F-2 | 26.0 |
| XYZ-B-H2-F-8 | 12.1 | XYZ-BR-H1-F-4 | 26.0 |
| XYZ-BR-H1 | 12.2 | XYZ-BR-H2-F-4 | 26.3 |
| XYZ-B-H1-F-2 | 12.6 | BOU-B-H2-F-4 | 26.8 |
| XYZ-B-H2-F-2 | 12.6 | BOU-B-H1-F-4 | 26.8 |
| XYZ-B-P2 | 12.8 | XYZ-B-H2-F-4 | 26.8 |
| BOU-B-H1-F-4 | 17.6 | BOU-BR-H2 | 27.1 |
| BOU-B-H2-F-4 | 17.6 | XYZ-B-H1-F-4 | 27.4 |
| **XYZ-B-S2** | 18.8 | XYZ-BR-H2 | 27.7 |
| BOU-BR-H1-F-8 | 19.5 | BOU-B-H1-L | 27.9 |
| <span style="color:red">BOU</span> | 20.0 | BOU-B-H2-L | 27.9 |
| BOU-BR-H1-F-4 | 21.0 | XYZ-B-H1-F-8 | 28.2 |
| BOU-B-H1-F-2 | 21.1 | XYZ-B-H2-F-8 | 28.2 |
| BOU-B-H2-F-2 | 21.1 | ARB-BR-H2-L | 28.3 |
| BOU-BR-H2-F-8 | 21.4 | ARB-BR-H1-L | 28.5 |
| BOU-BR-H1-F-2 | 22.3 | BOU-B-H1-F-8 | 28.6 |
| BOU-BR-H1 | 22.5 | BOU-B-H2-F-8 | 28.6 |
| BOU-BR-H2-F-4 | 23.7 | XYZ-BR-H1 | 29.2 |
| BOU-B-H1-F-8 | 24.4 | **BOU-B-S2** | 29.4 |
| BOU-B-H2-F-8 | 24.4 | XYZ-BR-H2-F-2 | 29.9 |
| BOU-BR-H2 | 25.3 | XYZ-B-H1-F-2 | 30.5 |
| BOU-B-H1 | 25.9 | XYZ-B-H2-F-2 | 30.5 |
| BOU-B-H2 | 25.9 | BOU-B-H1 | 32.6 |
| BOU-B-H1-L | 26.2 | BOU-B-H2 | 32.6 |
| BOU-B-H2-L | 26.2 | XYZ-B-H2 | 33.8 |
| XYZ-B-P1 | 26.8 | XYZ-B-H1 | 33.8 |
| BOU-BR-H2-F-2 | 28.0 | <span style="color:red">BOU</span> | 40.8 |
| ARB-BR-H1-L | 29.1 | <span style="color:red">XYZ</span> | 41.6 |
| XYZ-BR-P2 | 29.3 | ARB-BR-H1-F-8 | 42.9 |
| XYZ-BR-P1 | 29.5 | XYZ-B-P2 | 43.5 |
| **BOU-B-S2** | 30.5 | BOU-B-P2 | 44.0 |
| ARB-BR-H2-L | 31.4 | ARB-BR-H2-F-8 | 44.3 |
| <span style="color:red">ARB</span> | 49.5 | <span style="color:red">ARB</span> | 60.2 |

In conclusion, we were able to achieve our goal of finding similar solutions of good quality compared with design space exploration from scratch by using advanced routing techniques plus heuristics. The most successful configurations not only steer the solving to similar solutions but emphasize deciding similarity atoms first. The heuristic level modifier consistently performs well, followed by the factor modifier; meanwhile, different factor values did not have a significant impact. Strategies are applicable in limited capacity. For about half of the instance set, strategy s2 for binding could only obtain answers that performed well above the baselines and in some cases in the top 10. While the limitations are obvious, recall that strategies give guaranteed similarity in contrast to heuristics. Finally, preferences failed across the board. The more complex optimization led to less solutions found, and treating similarity on the same level as the other optimization criteria might lead to undesired results. One might try hierarchical Pareto optimization, but as of yet, our system is not capable of that.

## 7. Summary

We tackled the evolutionary system design (ESD) problem using answer set programming modulo difference constraints. Evolutionary system design consists of two steps:

1.  Perform design space exploration for a system synthesis problem and obtain and implement a high-quality solution;
2.  Perform design space exploration for a similar system synthesis problem while maximizing similarity to the previously obtained high quality solution.

This process allows for the swift finding of new design points that, first, are likely to have better quality because of the system synthesis problems having similar structure, and second, enhance the time-to-market for the implementation of a given design point as a product. We can use ESD for a variety of applications pertaining to embedded systems, such as successor generations, intra-generation variants, low-cost or high-performance variants, or functionality updates. We formalize all aspects of ESD, namely, the system synthesis problem, along with the design space exploration process, which consist of finding the Pareto front regarding the quality measures of latency, cost, and energy consumption, and similarity measures between two implementations.

We then presented AMT encodings that capture all these aspects. The system synthesis problem has been previously tackled with ASP [17–21]. This application is inherently hybrid as routing and conflict detection and resolution of messages are combinatorial in nature and require reachability, which can be easily handled with plain ASP. On the other hand, scheduling requires fine-grained timing involving linear constraints over integer variables, which we captured via difference constraints. A similar division of labor was also applied to other applications involving scheduling, e.g., train scheduling [22] and job shop scheduling [23]. Furthermore, we provided alternative message routing techniques for improving the solving performance. We introduced three classes of techniques for achieving similarity between implementations: strategies, preferences, and heuristics. Strategies restrict the search space and only allow for similar solutions. Preferences include similarity as an additional objective in Pareto optimization. Heuristics reorder the search space such that similar implementations are found first. To our knowledge, ESD has not been tackled to the extent presented in this paper with technologies such as ASP, satisfiability testing (SAT [24]), or similar combinatorial approaches. The underlying design space exploration, however, was addressed via different meta-heuristic techniques (e.g., [25,26]), exact methods such as integer linear programming (ILP) (e.g., [27,28]), and meta-heuristics combined with SAT (e.g., [29,30]). While meta-heuristics usually generate solutions faster for large instances, they might output the same or infeasible solutions repeatably; AMT, as with other exact methods, does not suffer from that drawback. Compared with other exact methods, though, ASP is uniquely suited to encode reachability, which is used for routing. As a matter of fact, reachability can be encoded natively and more succinctly in ASP compared with SAT. Finally, most of the previously mentioned methods only allow for single-objective optimization, while our ASP-based framework allows for a full Pareto

optimization. ESD can be compared with the minimal perturbation problem [31]; for instance, this problem has been solved using ASP for curriculum-based timetabling [32] and nurse re-rostering [33]. The main difference to our approach is that the quality is single-objective. We could formulate the ESD in terms of a minimal perturbation problem by having a hierarchical Pareto optimization, where the top level is bi-objective with stability defined with the Hamming distance and quality given by a vector of cost, latency, and energy consumption.

We systematically and empirically evaluated our techniques. Heuristics with advanced routing techniques proved to be the most successful, vastly outperforming DSE from scratch in terms of quality and similarity. Strategies were applicable for a subset of solutions. As strategies restrict the search space, the system synthesis problem might become unsatisfiable. If this is not the case though, strategies provide guarantees in terms of similarity that heuristics do not. Finally, preferences were unsuccessful due to the increased complexity of the Pareto optimization.

The sheer amount of techniques and variants that are involved in ESD leave open several avenues for future work. For instance, one can enhance the preference approach via a hierarchical Pareto optimization or have a more fine-grained similarity measure that also considers scheduling and similar mapping options.

**Author Contributions:** Conceptualization, C.H., L.M., K.N., T.S. and P.W.; methodology, C.H., L.M., K.N., T.S. and P.W.; software, L.M., K.N. and P.W.; writing—original draft preparation, C.H., L.M., K.N., T.S. and P.W.; writing—review and editing, T.S. and P.W. All authors have read and agreed to the published version of the manuscript.

**Funding:** This work was funded by the German Science Foundation (DFG) under grants HA 4463/4 and SCHA 550/11.

**Data Availability Statement:** System and benchmarks that were used can be found here: https://github.com/krr-up/asp-dse/releases/tag/v1.0.1 (accessed on 7 February 2023)

**Conflicts of Interest:** The authors declare no conflict of interest.

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
