# Peer review of "Evolutionary System Design with Answer Set Programming"

_algorithms, doi:10.3390/a16040179_

Round 1

Reviewer 1 Report

Evolutionary system design with ASP 

The article is devoted to an application of ASP, ASPmT if to be precise, within the realm of evolutionary system design. The paper is well structured. Evolutionary system design problem is well stated and accurately formalized. The authors describe several alternative implementations for this application that display distinct properties. They carefully analyze these variants within experimental evaluation. All and all the paper suits the venue very well and is ready to see the light.  

I have two somewhat more substantial remarks and then a small list of typo corrections. 

  1. The authors motivate well the importance of finding sensible solutions to Evolutionary system design problem. They also speak of their work as the only one in the context of ASP and other related paradigms. Yet, it is obvious that this problem has been studied and if nothing else then proprietary special-purpose solutions to this problem exist. Would it be possible to comment on this aspect in this paper. What is it that would motivate ASP solution to this problem that existing approaches cannot provide? How does ASP approach compare computationally to these? In other words, is there any way to add some reference points to where ASP solution to Evolutionary system design stands in comparison to its non-ASP peers. 

  1. Section 4.2.2 is truly difficult to follow. If nothing else it is really far from being self-contained. What I guessed is that predicate “preference” is special purpose predicate that a specialized propagator is devised for. Is this wild guess correct? What is the relation between preference predicate and pref mentioned at the closing of the section?  

Minor points 

  • line 213: starts->start 

  • Line 281: put mathematic expression into equation style 

  • Line 284: explicitly -> explicit; if-> of 

  • Line 304: add and at the end of the 1st bullet 

  • Line 313: is “identical” a term? It was not defined 

  • Line 378: drop “actual” 

  • The paragraph starting at line 491 mixes the notation and names of predicates used within a program and conditions listed in the definitions. It would help for this description to be more precise so that when the lines of the programs are being referred to it were explained how predicate names used within this code encode the requirements of the conditions. 

  • Line 963: unsatisfiability-> unsatisfiable 

  • As far as experiments: they were nicely summarized within the narrative. Yet, I found difficult to navigate the tables in the form they are. How do columns align? 

Author Response

Thank you for your comments!

In regards to the usefulness of ASP compared to other technologies, we added the following paragraph in the Summary section:
"The underlying design space exploration, however, was addressed via different meta-heuristic techniques (e.g.~\cite{thopim13a,felapisctu10a}),
exact methods like Integer Linear Programming (ILP) (e.g.~\cite{luglhate08a,khrosa16a}),
and meta-heuristics combined with SAT (e.g. \cite{nehagl16a,scluhate06a})
While meta-heuristics usual generate solutions faster for large instances,
they might output the same or infeasible solutions repeatably.
AMT, as other exact methods, do not suffer from that drawback.
Compared to other exact methods, though, ASP is uniquely suited to encode reachability which is used for routing.
As a matter of fact, reachability can be encoded natively and more succinctly in ASP compared to SAT.
Finally, most of the previously mentioned methods only allow for single-objective optimization while our ASP-based framework allows for full Pareto optimization."
The advantages and disadvantages of ASP for DSE directly propagate to ESD.

You are correct, the preference predicate is a special predicate that is known to the Pareto optimization system. We added the following to make this more clear:
"Preference definitions rely on atoms over predicates \lstinline|preference/2| and \lstinline|preference/5|.
Additionally, rules are provided capturing atoms that the preferences relate to via atoms over predicate \lstinline|holds/2|.
These predicates are known to the Pareto optimization system and used to communicate the desired preferences."

The symbols over pref/3 however, are added to the answer sets after a solution was found to indicate the value of the preferences. For instance, pref(cost,sum,12) would indicate that the preference named cost of type sum has the value 12.
In the paper, we explain this with the sentences:
"Similar to \clingodl,
our Pareto optimization framework adds symbols to the answer set describing the quality of the solution.
Specifically, we add symbols of the form \lstinline[mathescape]|pref($n$,$t$,$v)$|,
where $n$ is the name,
$t$ the type,
and $v$ the objective value."

All minor points where addressed.

To clarify the encoding explanation, we added:
"In the following, we argue on the ground level and with the mathematical entities of the system synthesis problem.
That is, every \lstinline|T| and \lstinline|T'| in the encoding is instantiated by all tasks $\{t,t'\}\subseteq\alltasks$,
as well as every \lstinline|C| and \lstinline|C'| is instantiated with all communications $\{c,c'\}\subseteq\alldependencies$.
"

Finally, we changed the column alignment of the tables to improve readability.

Reviewer 2 Report

Paper is written clearly and logically. It is clear that author continue on their previous research results in this area. Paper is well structured: authors provide sufficient introduction and motivation, theoretical foundations. The core of the paper are sections 4 and 5. Experiments are documented and discussed in sect. 6.

There are no factual nor logical mistakes. All experiments are supported by results.

I recommend a proofreading and accepting the paper.

Author Response

Thank you for the review! We further proofread and improved the paper.